# Divergent abiotic spectral pathways unravel pathogen stress signals across species

P. J. Zarco-Tejada [1,2 ✉], T. Poblete[1], C. Camino[3], V. Gonzalez-Dugo[2], R. Calderon[4], A. Hornero[5,2], R. Hernandez-Clemente[5], M. Román-Écija[2], M. P. Velasco-Amo[2], B. B. Landa[2], P. S. A. Beck[3], M. Saponari[6], D. Boscia[6] & J. A. Navas-Cortes [2]

Plant pathogens pose increasing threats to global food security, causing yield losses that exceed 30% in food-deficit regions. *Xylella fastidiosa* (*Xf*) represents the major transboundary plant pest and one of the world's most damaging pathogens in terms of socioeconomic impact. Spectral screening methods are critical to detect non-visual symptoms of early infection and prevent spread. However, the subtle pathogen-induced physiological alterations that are spectrally detectable are entangled with the dynamics of abiotic stresses. Here, using airborne spectroscopy and thermal scanning of areas covering more than one million trees of different species, infections and water stress levels, we reveal the existence of divergent pathogen- and host-specific spectral pathways that can disentangle biotic-induced symptoms. We demonstrate that uncoupling this biotic–abiotic spectral dynamics diminishes the uncertainty in the *Xf* detection to below 6% across different hosts. Assessing these deviating pathways against another harmful vascular pathogen that produces analogous symptoms, *Verticillium dahliae*, the divergent routes remained pathogen- and host-specific, revealing detection accuracies exceeding 92% across pathosystems. These urgently needed hyperspectral methods advance early detection of devastating pathogens to reduce the billions in crop losses worldwide.

[1] School of Agriculture and Food (SAF-FVAS) and Faculty of Engineering and Information Technology (IE-FEIT), University of Melbourne, Melbourne, VIC, Australia. [2] Instituto de Agricultura Sostenible (IAS), Consejo Superior de Investigaciones Científicas (CSIC), Avda. Menéndez Pidal s/n, 14004 Córdoba, Spain. [3] European Commission, Joint Research Centre (JRC), Ispra, Italy. [4] Plant Pathology and Plant-Microbe Biology Section, School of Integrative Plant Science, Cornell AgriTech, Cornell University, Geneva, NY, USA. [5] Department of Geography, Swansea University, Swansea SA2 8PP, UK. [6] CNR, Istituto per la Protezione Sostenibile delle Piante, Bari, Italy. ✉email: pablo.zarco@unimelb.edu.au

Each year, plant pathogens cause an estimated 16% production loss globally, a number that has not significantly diminished over the last 40 years despite increased pesticide use[1]; in food-deficit regions, yield losses due to plant pathogens can exceed 30%[2]. Climate change[3] and global trade[4] are escalating the damage to agricultural production and food security caused by invasive species[5] and both emerging and reemerging pathogens responsible for plant diseases[2,6,7]. Globally, agricultural and forestry production is threatened by the rapid expansion of the Ug99 race and other new races of stem rust (*Puccinia graminis* f.sp. *tritici*) infecting wheat (*Triticum aestivum*) in Africa, the Middle East and Asia[8], as well as pathogens including the tropical race 4 (TR4) of *Fusarium oxysporum* f.sp. *cubense* on banana (*Musa acuminata*) Cavendish cultivars in Southeast Asia[9], Citrus canker (*Xanthomonas axonopodis* pv. *citri*)[10], and Citrus greening (*Candidatus Liberibacter* spp.) in the Americas[11], and *Xylella fastidiosa* within Europe infecting olive (*Olea europaea*)[12], almond (*Prunus dulcis*)[13], and grape (*Vitis vinifera*)[14].

Of these pathogens, *Xylella fastidiosa* (*Xf*)[15] is arguably the greatest threat worldwide, causing enormous socioeconomic and environmental losses[3,16]. It can infect over 550 plant species[17] and has been identified as a major transboundary plant pest[18]. In the Americas, *Xf* is associated with diseases of grapevine, almond, coffee (*Coffea* spp.) and citrus, causing sizable economic losses[19]. Its recent invasion into some European countries is devastating olive and almond groves, with both economic and environmental consequences. In a hypothetical scenario modelling massive spread throughout Europe, *Xf* was projected to disrupt agriculture production to the level of up to €5.2 billion of losses per year in the olive sector alone[20]. Outside America and Europe, the spread of this pathogen in Asia via Iran[21] and Taiwan[22], and its 2019 identification in Israel has intensified international calls to contain this global *Xf* epidemic.

The development of robust large-scale plant scanning methods will be key to successfully monitor detrimental crop pathogens and assist in their timely eradication or optimise containment measures[23]. Advanced imaging spectroscopy is the only large-scale method that allows early detection of infectious plant diseases, i.e. when symptoms are not visible yet but spread of the pathogen can occur[24]. Hyperspectral imaging has been recently used to detect, for example, rice sheath blight[25], tobacco mosaic virus[26], late blight, target and bacterial spots[27], spotted wilt virus in tomato[28], phytophthora-induced decline[29], verticillium wilt and the olive quick decline syndrome[30]. However, a major limitation of advanced hyperspectral, thermal scanning and radiative transfer methods in plant health monitoring is that the subtle physiological alterations caused by a disease reflect changes in plant physiological state, such as stomatal regulation[31] and the coupled chlorophyll fluorescence-photosynthesis-transpiration dynamics[32], which are all commonly modulated by both biotic and abiotic confounding factors. Revealing distinct spectral fingerprints associated with biotic- vs. abiotic-stress conditions is thus of paramount importance for large-scale remote detection efforts of early disease infection symptoms that occur in the context of natural physiological variability (e.g., due to water deficit or nutrient deficiencies) commonly found even in irrigated croplands.

In this study, we successfully disentangled biotic stress caused by vascular system-invading pathogens from abiotic stress imposed by water limitation by revealing distinct spectral pathways associated with the physiological alterations detected through imaging spectroscopy and thermal data. We carried out airborne campaigns scanning over one million infected and healthy trees across seven regions in Italy and Spain between 2011 and 2019 (Fig. 1). To elucidate the host-specific spectral fingerprints for *Xylella fastidiosa* infections, we flew over officially designated *Xf* outbreaks affecting olive and almond fields. In olive, we also investigated whether we could distinguish between the effects of distinct xylem-limited pathogens that cause similar physiological symptoms. We evaluated whether *Xf*-associated biotic–abiotic spectral fingerprints were distinct from those detected for *Verticillium dahliae* (*Vd*), the xylem-invading fungus that causes Verticillium wilt, the most devastating soilborne disease infecting olive trees worldwide[33]. Notably, these two distinct xylem-limited pathogens increase resistance and eventually block water flow through the vascular system[34]. This collapse in water flow reduces transpiration and induces water stress, thus causing analogous symptoms that also can be confounded with abiotic stress[35]. To assess the existence of divergent *Xf*- and *Vd*-induced biotic–abiotic spectral alterations, we analysed a subset of ca. 380,000 healthy trees encompassing agricultural fields grown under variable water stress levels for both host species. We used these data to monitor (i) how the *Xf* pathogen affected two different species (almond vs. olive), and (ii) how one species (olive) responded to infection by two different xylem-limited pathogens (*Xf* vs. *Vd*). Our aim was to evaluate the robustness of distinct spectral traits to detect the biotic stress-induced symptoms, comparing across species and pathogens, while disentangling their specific spectral alterations from those caused by abiotic stress-induced dynamics.

## Results and discussion

**Biotic stress-induced spectral alterations across species and pathogens.** Our analysis of high-resolution airborne hyperspectral and thermal images collected over *Vd* (Fig. 2a) and *Xf* (Fig. 2b–d) outbreaks showed that infection-induced physiological alterations led to changes in biotic stress-sensitive spectral traits that were common between host species, while other traits deviated between the plant species and appeared to be host-specific (Fig. 2b vs. c). In both host species, *Xf* infection altered spectral plant traits related to stomatal conductance dynamics as the infection progressively blocked xylem vessels and thus reduced transpiration[36]. Lower transpiration rates also raise the overall tree canopy temperature, as measured by the thermal crop water stress index, CWSI[37], which is accompanied by a reduction in photosynthesis observed through solar-induced fluorescence emission signal (SIF), and alterations in the dynamics of the xanthophyll pigment cycle (for which the normalised Photochemical Reflectance Index, $PRI_n$, provides a proxy) (see Supplementary Table 1 for a complete list of spectral plant traits). Remarkably, our results illustrated species-specific spectral traits altered by *Xf*-induced stress: the blue-region spectral trait NPQI, which is related to chlorophyll-phaeophytin degradation[24], and anthocyanin content (Anth.) were of limited importance under *Xf* infection in almond trees yet extremely relevant to detect *Xf* infection in olive trees (Fig. 2b, c).

We obtained these results through a multilayered functional plant-trait scheme[24] derived from the inversion of a physical radiative transfer model and a machine learning (ML) algorithm[30], applied here for the first time across two different host species. The numerous visible and near-infrared (VNIR) spectral indices initially calculated (Supplementary Table 1) were reduced by a multicollinearity analysis based on the variance inflation factor (VIF). The latter enabled the enhanced contribution of the thermal trait (CWSI), the solar-induced fluorescence calculated at 760 nm ($SIF_{@760}$), and the model-estimated traits such as the leaf biochemical constituents and the canopy structural parameters on the disease detection. To make the results comparable across species and pathogens, the obtained importance for each spectral trait was normalised by the highest importance of each pathogen/species within each ML model (see

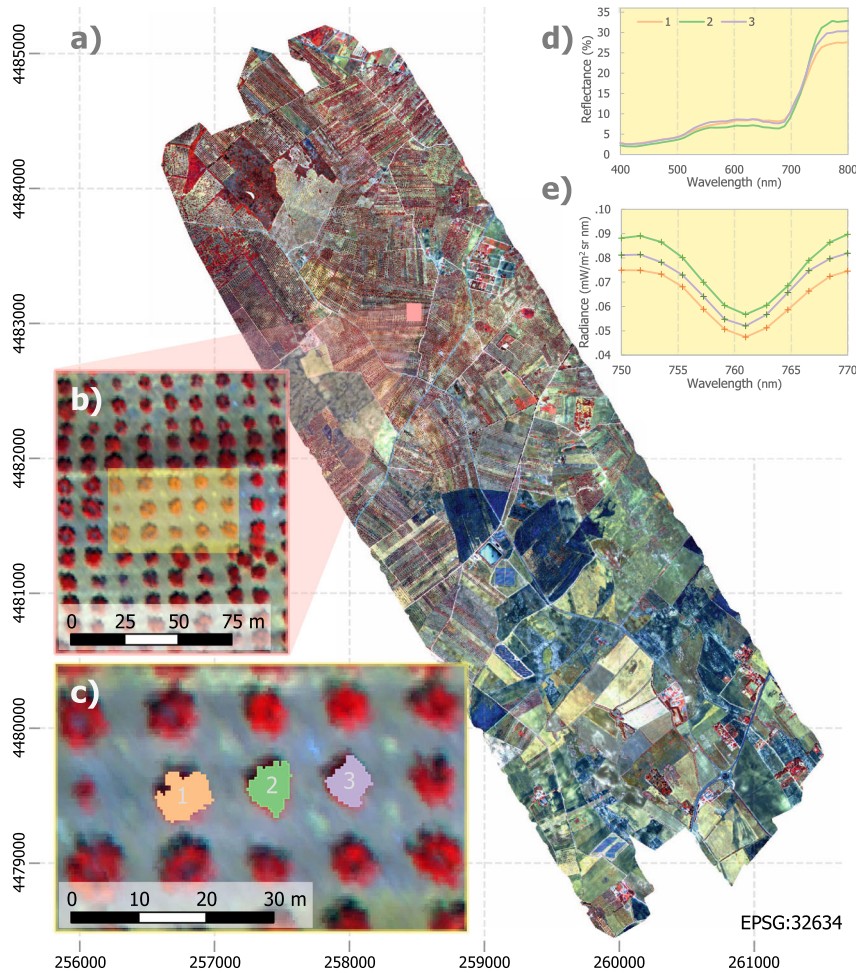

**Fig. 1 High-resolution airborne hyperspectral image acquired over one of the Apulia *Xylella fastidiosa* (*Xf*)-infected areas.** Similar datasets were collected from all *Xf* and *Verticillium dahliae* (*Vd*) outbreaks used for the analyses carried out in this study. **a** mosaic covers 1450 ha at 40 cm resolution collected with 260 spectral bands. **b** individual trees could be identified on the images and properly located during field work. Tree-crown segmentation algorithms (**c**) were used for the selection of pure vegetation pixels. Extracted reflectance (**d**) and radiance spectra (**e**) were used to calculate spectral indices, plant traits by radiative transfer model inversion, and solar-induced fluorescence (SIF) used as inputs for the disease-detection models.

Methods section for detailed description). This approach revealed, on a common scale, how important individual spectral plant traits were in the overall response of the two host species studied here to biotic and abiotic stressors. We describe in detail how spectral plant traits are expressed in *Xf*-infected trees depending on the tree water status (Fig. 2c vs. d). We show that anthocyanins are not a sensitive indicator of *Xf* infection in almond trees irrespective of their water stress levels (Fig. 2c), while *Xf* infection evoked an NPQI response only in well-watered almond trees (Fig. 2d). Importantly, we observed that the two xylem-limited pathogens (*Xf* and *Vd*) infecting olive trees left distinct spectral plant-trait fingerprints on their hosts (Fig. 2a vs. b). We observed that a blue-region B spectral plant trait was expressed in *Vd*- but not *Xf*-infected plants, and that the most sensitive indicators of *Xf* infection in olive after CWSI, namely NPQI, chlorophyll fluorescence and PRI$_n$-xanthophyll spectral traits, were relatively uninformative for *Vd* infection. By contrast, CWSI and anthocyanin contents were sensitive spectral traits to both *Xf* and *Vd* infection in olive. These results demonstrate that the sensitivity of specific spectral plant traits is a function of the nature of the biotic stressor: when a pathogen (*Xf*) infects multiple host species (olive vs. almond) (Fig. 2b vs. c) and when different xylem-limited pathogens (*Xf* vs. *Vd*) infect the same host species (olive) (Fig. 2a vs. b).

The spectral changes revealed in tree populations experiencing biotic stress in the form of *Xf* or *Vd* infections by means of imaging spectroscopy, thermal indicators and radiative transfer methods, are consistent with fundamental leaf-level physiological processes. Infected vegetation accumulates photoprotective compounds such as phenolics[38], flavonoids and carotenoids (C$_{x+c}$) that also act against plant pathogens[39]. In the case of *Xf* infections, laboratory assays[40] and spectral analyses have demonstrated an increase in leaf temperature and anthocyanins content, as well as a reduction of chlorophyll fluorescence[24] that is accompanied by a degradation of photosynthetic pigments. Our optical measurements, taken in situ from leaves of *Xf*-infected olive and almond trees, confirmed the sensitivity of the spectral plant traits identified from airborne imaging spectroscopy (Figs. 2e to 2j. Consistent with our tree crown-level image analyses, we observed that leaf temperature (Fig. 2e), fluorescence emission (Fig. 2h) and xanthophyll-related spectral traits (Fig. 2i) were sensitive to *Xf* infection across species. By contrast, NPQI at the leaf level was only sensitive under well-watered conditions (Fig. 2f). At the same time, our results demonstrate at two different scales (leaves and tree crowns through airborne imaging spectroscopy) that the sensitivity of specific spectral indicators induced by biotic stress is modulated by the water status of the infected vegetation.

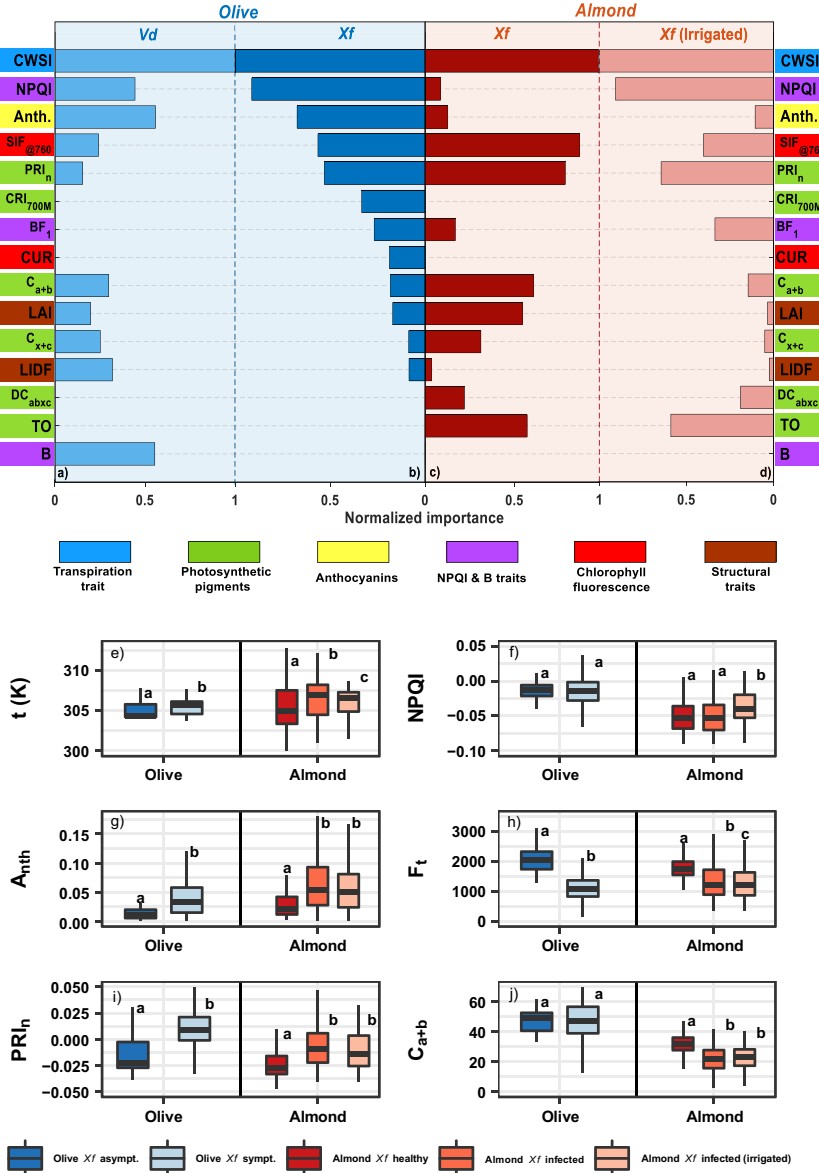

**Fig. 2 Importance of spectral traits to detect *Xf*- and *Vd*-infection symptoms. a–d** Normalised importance of hyperspectral and thermal plant traits retrieved from the pool of spectral plant traits used to detect *Verticillium dahliae*, *Vd*- (**a**) and *Xylella fastidiosa*, *Xf*-induced infection symptoms (**b–d**) across olive (**a**, **b**) and almond (**c**, **d**) trees. For reference, the full list of spectral plant traits is available in the Supplementary Table 1. The importance analysis was carried out using a balanced training dataset obtained from $n = 1878$ (**a**), $n = 7296$ (**b**), $n = 4048$ (**c**), $n = 2680$ (**d**) trees by the permutation of out-of-bag (OOB) predictor methodology. The importance of each spectral trait was normalised by the highest importance obtained for each disease/species within each ML model. **e–j** Analysis of spectral plant traits measured in the field from asymptomatic vs. *Xf*-infected olive and almond leaves. **e** Temperature at midday (t, $n = 1797$ leaf samples). **f** Normalised phaeophytinization index-based spectral trait (NPQI, $n = 1457$ leaf spectral samples). **g** Anthocyanins (Anth, $n = 1318$ leaf samples). **h** Steady-state leaf chlorophyll fluorescence ($F_t$, $n = 2784$ leaf samples). **i** Normalised xanthophyll cycle dynamics index (Photochemical Reflectance Index [$PRI_n$], $n = 1457$ leaf spectra samples). **j** leaf chlorophyll content ($C_{a+b}$, $n = 2584$ leaf samples). Statistical analyses were carried out by a one-way non-parametric ANOVA (Kruskal-Wallis test) followed by a Wilcoxon post-hoc test (two-sided) with Bonferroni correction to examine significant differences at $p$-value < 0.05 between the leaf groups for each species. Severity levels with the same letter are not significantly different ($p$-value $\geq 0.05$). Exact $p$-values are provided in the repository indicated in the Data availability and Code availability sections that contain the source data and code. The horizontal black line in the boxplots displays the median, and the top and bottom horizontal lines represent the 75th and 25th percentiles, respectively. Whiskers display the lower and upper limits of interquartile ranges (Q ± 1.5xIQR).

**Disentangling biotic–abiotic confounding symptoms on spectral traits.** A major limitation towards the wide use of the alterations induced by biotic stress that can be detected by imaging spectroscopy is how intrinsically entangled they are with physiological changes stemming from abiotic stressors that are routinely experienced in agricultural fields. In the absence of sources of biotic stress, the restriction of canopy growth under

suboptimal water or nutrient levels is generally associated with stomatal closure[41] and chlorosis[42], thus imposing both water and nutritional stress due to reduced uptake by the root system. Similarly, photosynthetic rate diminishes in response to both stomatal and non-stomatal effects[43], with a contrasting level of recovery after the release of stress[44]. In practice, these coupled biotic–abiotic physiological effects observed as broadly similar

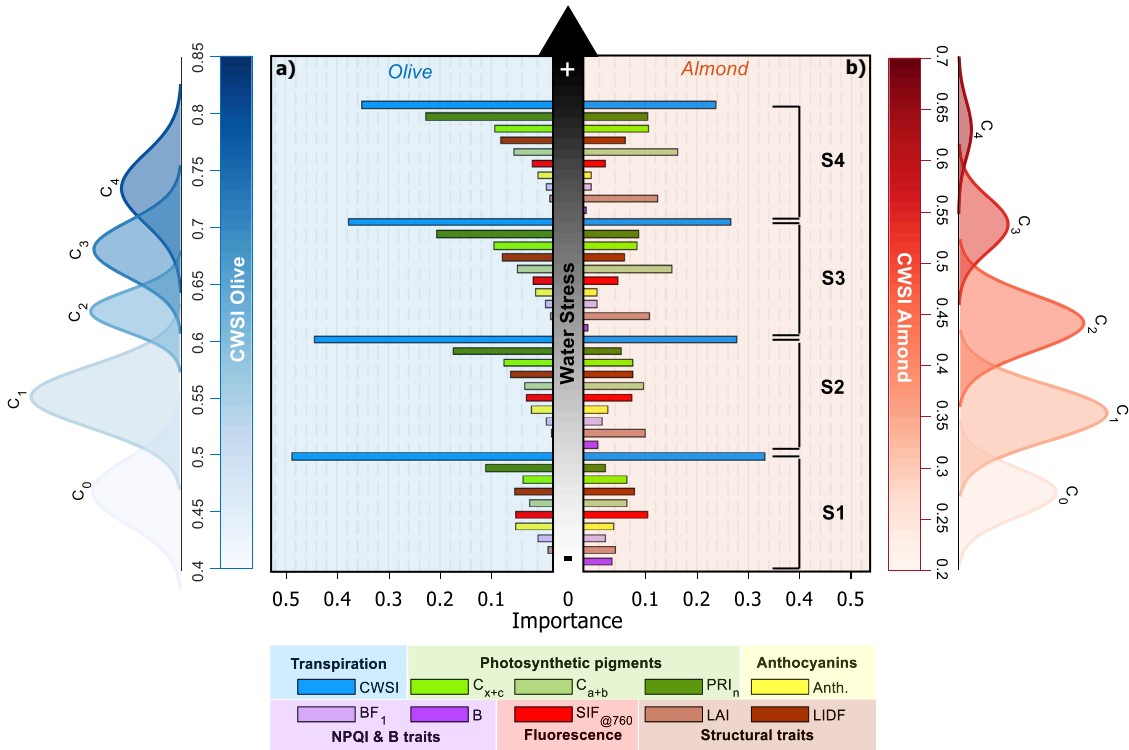

**Fig. 3 Importance of spectral traits to detect abiotic-induced water stress symptoms.** Sensitivity of plant spectral traits calculated from hyperspectral and thermal imagery from trees under increasing abiotic stress (caused by decreasing water stress levels, from S1 to S4) across olive (**a**) and almond (**b**) trees. Analyses were carried out via clustering by comparing non-stressed trees ($C_0$) vs. trees with rising CWSI levels ($C_1$ to $C_4$). Clustering was performed based on CWSI by following a modified three-sigma rule, where $C_0$ consists of trees in the lowest 10th percentile. Clusters $C_1$, $C_2$, $C_3$, and $C_4$ were set to include only the trees with CWSI values above the 10th and below the 68th percentile ($C_1$), between the 68th and 85th percentiles ($C_2$), above the 85th and below the 95th percentile ($C_3$), and above the 95th percentile ($C_4$). For (**a**), the total number of trees was $n = 488$ ($C_0$), $n = 3,066$ ($C_1$), $n = 1,090$ ($C_2$), $n = 618$ ($C_3$) and $n = 222$ ($C_4$). For **b**, the total number of trees was $n = 390$ ($C_0$), $n = 1,776$ ($C_1$), $n = 1,248$ ($C_2$), $n = 214$ ($C_3$) and $n = 24$ ($C_4$). The importance of each predictor on the classification was assessed by the permutation of out-of-bag (OOB) predictor methodology applied as a random forest algorithm.

spectral fingerprints have thus far hindered the successful large-scale remote detection of infected trees.

To unravel these confounding spectral alterations induced by biotic and abiotic stressors, we implemented a feature-weighted ML algorithm based on the methodology proposed by Liu and Zhao[45]. The feature-weighted layer that we developed accounts for the importance of the most sensitive spectral traits to detect *Xf*- and *Vd*-infected olive trees. Considering the predictions of the feature-weighted ML model, we evaluated the uncertainty and the performance of detection models for infection by *Xf* and *Vd* based on spectral plant traits in terms of their overall accuracy (OA) and kappa coefficients ($\kappa$). We then validated our detection models against molecular diagnostic assays performed in the field or on field samples: conventional polymerase chain reaction assay (PCR), quantitative PCR (qPCR) and recombinase-polymerase-amplification (RPA), and by visual inspections carried out in outbreak areas. Our feature-weighted models, which at this stage account only for the biotic stress-induced variability in the spectral traits yielded OA = 84% ($\kappa = 0.68$) for *Xf* detection of infection in almond ($n = 4048$ trees), and OA = 77% ($\kappa = 0.43$) and OA = 75% ($\kappa = 0.49$) for *Xf* and *Vd* detection of infection in olive, respectively ($n = 7296$ and $n = 1852$ trees, respectively).

Despite obtaining classification accuracies exceeding 75%, we also noticed a large number of trees classified with high uncertainty based on the classification probabilities of the featured-weighted ML algorithm. These contained most of the trees misclassified as infected by the ML algorithms but showing

no visual symptoms—thus considered as false positives (FP) for infection— and trees classified as not infected but showing visual symptoms—thus false negatives (FN) for infection—with a total of 65%, 72% and 50% uncertain trees for *Xf* and *Vd* in olive, and *Xf* in almond, respectively. We hypothesised that these large uncertainties in the detection of *Xf* and *Vd* infection symptoms across species might be due to the role of the physiological responses that are commonly triggered by both biotic and abiotic stressors, thereby causing similar reductions in leaf water potential, $CO_2$ assimilation, and stomatal conductance[36] along with decreased chlorophyll fluorescence and changes in pigment concentrations[24].

**The importance of water stress quantification in reducing vascular disease-detection uncertainty.** We disentangled the changes in spectral plant traits caused by biotic (*Xf* and *Vd* infection) and abiotic stressors by analysing a temporal series of airborne imaging spectroscopy and thermal imagery acquired in areas free of pathogens yet experiencing variable water status levels. We flew over ca. 380,000 olive and almond trees across three geographical regions with our airborne imaging sensors over two summer growing seasons. This multitemporal dataset enabled the identification of individual trees that showed consistent and sustained water status levels across seasons. The temporal approach allowed clustering of trees with different stress levels over a long period of time (i.e., two growing seasons) rather

than focusing the analysis on short-term water stress conditions potentially due to transitory environmental effects or irrigation system malfunctions in each orchard under study. Thus, the multiyear dataset improved the selection of trees consistently experiencing sustained water stress. The scanned fields contained irrigated groves that followed current best practices and water availability, with recorded values for stem water potential ranging from –1.7 to –1.9 MPa over the course of the season for almond trees, and –2.0 to –2.9 MPa in olive trees. Applying a modified three-sigma rule approach to these multitemporal datasets, we used the thermal-based transpiration trait CWSI calculated across years to cluster the ca. 50,000 scanned olive and almond trees into groups based on percentiles (10th, 68th, 85th and 95th) corresponding to non-stressed trees ($C_0$) vs. clusters with increasing water stress levels ($C_1$ to $C_4$) (Fig. 3). We then used the multi-layered radiative transfer model inversion and the weighted-based ML algorithm to assess the dynamics of the respective spectral traits as a function of increasing levels of water stress (indicated as S1 to S4 stress levels in Fig. 3).

We identified a set of spectral plant-trait indicators that was consistently sensitive to water limitation in both species and in the absence of biotic stress (Fig. 3). The trends of these abiotic-induced indicators deviated from those observed under $Xf$-infection conditions, which moreover differed across the two species (Fig. 2b vs. c) as well as from those seen in the same host species for the two pathogens ($Xf$ vs. $Vd$) (Fig. 2a vs. b). The most important spectral trait across all water-limited abiotic stress levels was CWSI, which is consistent with the reduced stomatal conductance and transpiration of the plants, resulting in rising leaf temperatures, followed by an alteration of xanthophyll cycle dynamics ($PRI_n$). Remarkably, our results show that, as water stress increased (Fig. 3, S1 to S4 water stress levels), the relative importance of transpiration-related spectral indicators such as CWSI decreased, while that of physiological traits related to plant pigments and tree structure increased. In almond (Fig. 3b), adaptive mechanisms to severe water stress include defoliation to prevent desiccation[46]. In sharp contrast, olive trees (Fig. 3a) predominantly control water loss via transpiration by strict stomatal regulation[47]. These species-specific adaptive mechanisms led to distinct trends for the leaf area index (LAI) and chlorophyll content ($C_{a+b}$) spectral traits measured in both species (Fig. 3a vs. b). We show that the importance of the spectral indicators CWSI, anthocyanins content and SIF exhibit an inverse correlation with water stress levels. We also discovered critical information to help disentangle the detection of biotic and abiotic stress: several highly sensitive spectral plant traits identified in the context of biotic stress responses in olive (Fig. 2) showed either no sensitivity (such as the NPQI spectral trait) or only a weak sensitivity (Anth.) to various abiotic stress conditions in both tree species (Fig. 3a, b).

**Divergent abiotic–biotic spectral pathways unravel pathogen-specific detection traits**. We used the specific spectral traits linked to the biotic stress imposed by $Xf$ and $Vd$ infection to determine, for each pathosystem, which indicators deviated from the abiotic stress response (Fig. 4a, b, indicated with black stars), thereby unravelling the biotic–abiotic uncertainty affecting the screening models. Our results indicated that airborne-quantified fluorescence $SIF_{@760}$ is effective in distinguishing between $Xf$ infection and abiotic stress in both species. We also identified species-specific spectral traits that accomplished the same distinction: the pigment degradation-related NPQI spectral trait and anthocyanins content were specific to olive trees (Fig. 4a), while the xanthophylls-related trait $PRI_n$ and chlorophyll $a + b$ were almond-specific (Fig. 4b). However, these traits diverged when

irrigated almond trees were analysed separately (Fig. 4c), indicating that NPQI constitutes a distinct marker trait for $Xf$ infection in almond trees only under non-water-limited conditions. Interestingly, results from the $Vd$ dataset identified spectral traits NPQI and B as specific markers for $Vd$ infections independently of abiotic stress status (Fig. 4d), demonstrating the importance of the blue spectral region for disentangling pathogen-induced stress from abiotic stress.

We then applied these newly revealed spectral plant trait fingerprints to re-evaluate in detail the trees that were classified with high uncertainty earlier by the ML algorithm (38% of olive and 17% of almond trees) and to reassess the results against the molecular diagnostic assays for false positive (FP) and false negative (FN) trees misclassified for each species. Using the spectral traits that distinguished between biotic and abiotic stressors as input for a spectral clustering algorithm, we disentangled the biotic–abiotic uncertainty, reducing the percentage of misclassified trees to 6.5% and 6.6% for olive and almond trees, respectively. These results thus supported our hypothesis that the original misclassification of trees was predominantly caused by uncertainty related to confounding biotic–abiotic physiological disturbances, and that the newly identified biotic–abiotic spectral fingerprints significantly reduced uncertainty across species and pathogens. Accounting for species- and pathogen-specific spectral traits (Fig. 4e, f displayed for $Xf$ in olive and almond) greatly improved model performance for all datasets comprising both species and both pathogens. Model accuracies for $Xf$ in almond reached OA = 94% ($\kappa = 0.87$), which we validated against qPCR results ($n = 265$), up from the original OA = 83% ($\kappa = 0.65$), while we achieved OA = 92% ($\kappa = 0.83$), up from the original OA = 62% ($\kappa = 0.25$) (qPCR $n = 77$) in olive trees. In the case of $Vd$-infected trees, we achieved OA = 93% ($\kappa = 0.87$), up from the original OA = 75% ($\kappa = 0.49$) (visual inspection, $n = 1852$).

**Towards global-scale monitoring of threatening plant diseases using imaging spectroscopy**. The work presented here demonstrates that potentially confounding symptoms of biotic and abiotic stress can be distinguished for particular host plant species. Our analyses of the most comprehensive high-resolution imaging dataset of pathogen-specific hyperspectral traits compiled so far show, for the first time, the existence of host- and pathogen-specific spectral plant responses that diverge between biotic and abiotic stresses. Our work goes beyond current knowledge, accurately detecting harmful xylem-limited pathogens across host species.

Global warming and international trade are exacerbating risks related to emerging and reemerging pathogens threatening agriculture. At the same time, world food production needs to increase by 50% over the next 30 years to feed a growing global population, despite decreasing arable land and climate disruption[48]. With yield losses due to pathogens exceeding 30% in regions where food security is critical, the development of technologies for large-scale early detection of outbreaks is crucial. A global plant disease monitoring framework will require collaboration across disciplines, including remote sensing, physics, artificial intelligence, engineering and sensor development, and space and drone industries, interacting closely with plant pathology, physiology, and agronomy.

The analytical approach introduced here provides a transferrable framework to disentangle pathogen-induced stress from abiotic dynamics in a range of species, which is critical for the development of global disease-detection models. Detecting the coexistence of both factors is also fundamental to evaluate the evolution and treatment of the plant, either to adapt the treatment in case there is only one stress factor, or to control the interaction of biotic and abiotic stresses. For

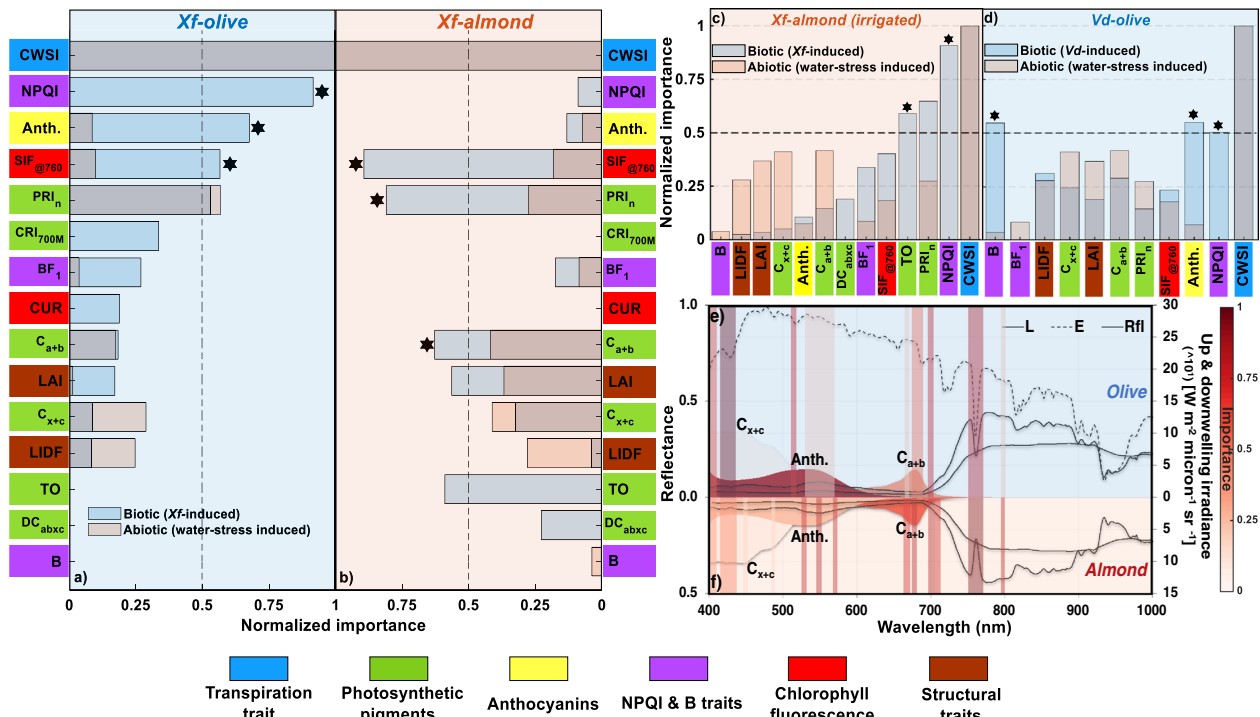

**Fig. 4 Importance of spectral plant traits for *Xf*- and *Vd*-detection across species under simultaneous biotic and abiotic stress.** Spectral plant traits that diverge under biotic and abiotic stress are indicated with black stars for *Xylella fastidiosa* (*Xf*) infection, in olive (**a**) and almond trees (**b, c**) and for *Verticillium dahliae* (*Vd*) infection in almond trees (**d**). The particular case of *Xf*-infected almond trees grown under non-water-limited conditions is shown in **c**. Importance of the different spectral regions for *Xf* detection in olive (**e**) and almond (**f**) trees showing the spectral radiance (L), irradiance (E) and reflectance (Rfl). The importance of each predictor was obtained by the permutation of out-of-bag (OOB) predictor methodology when classifying both biotic and abiotic stress-induced conditions using a random forest algorithm. The importance of each spectral trait was normalised by the highest importance obtained for each pathogen/species. Statistical analysis was carried out using a balanced training set obtained from the indicated number of trees: $n_{biotic} = 7296$ and $n_{abiotic} = 5484$ (**a**); $n_{biotic} = 4048$ and $n_{abiotic} = 3652$ (**b**); $n_{biotic} = 2680$ and $n_{abiotic} = 3652$ (**c**); $n_{biotic} = 1878$ and $n_{abiotic} = 5848$ (**d**).

example, drought can play a key role in the development of plant diseases[49,50]. The applicability of this framework to other pathotypes will require further considerations; for individual plant diseases, it will depend on the degree of divergence of the spectral pathways induced by the coupled biotic–abiotic stress-related physiological alterations for each species. We expect results to further improve for non-xylem-limited pathogens that cause physiological responses uncoupled from the abiotic dynamics of water stress. Widespread use will require further developments of technology readiness; a critical limitation for the operational application of these methods lies in the need for high-spatial resolution hyperspectral and thermal imaging (i.e., at sub-metre resolutions), a technology currently available only from drones at small scale, and from manned airborne platforms such as the ones used here. Future hyperspectral sensors on board satellites or high-altitude drones may enable systematic data collections with imaging spectroscopy at sub-meter resolution data, and, when combined with analytical frameworks, permit the real-time monitoring of diseases and abiotic stresses at global scales.

## Methods

**Airborne hyperspectral and thermal image acquisition.** We scanned over one million olive and almond trees between 2011 and 2019 with an airborne imaging spectroscopy and thermal imaging facility targeting infected and healthy trees in seven different regions located in Apulia (Italy), Majorca (Balearic Islands, Spain), Alicante, Cordoba and Seville (mainland Spain). In olive groves, over 200,000 and 372,000 trees were imaged from *Xf* and *Vd* outbreaks, respectively. In almond groves, we scanned over 132,000 trees from *Xf* outbreaks in Alicante and Majorca. To evaluate the effects induced by abiotic stress on spectral plant traits, we surveyed over 370,000 healthy trees (outside the outbreak areas) comprising olive and almond species subjected to a wide range of water stress conditions.

We surveyed these areas with airborne hyperspectral and thermal cameras on board a manned aircraft flying at 500 m altitude above ground, yielding 40 cm and 60 cm spatial resolution, respectively. We used a hyperspectral camera (VNIR model, Headwall Photonics, Fitchburg, MA, USA) collecting 260 bands in the 400–885 nm region at 1.85 nm/pixel and 12-bit radiometric resolution with a frame rate of 50 Hz. With this spectral configuration, we captured imagery at 6.4 nm full-width at half-maximum (FWHM) bandwidth and obtained an instantaneous field of view (IFOV) of 0.93 mrad and an angular field of view (FOV) of 49.82 deg with an 8 mm focal length lens. The hyperspectral sensor was radiometrically calibrated in the laboratory using an integrating sphere (CSTM-USS-2000C Uniform Source System, LabSphere, North Sutton, NH, USA). At the time of flight, we measured aerosol optical thickness at 550 nm using a Sunphotometer (Microtops II S model 540, Solar LIGHT Co., Philadelphia, PA, USA). We then applied the resulting atmospheric correction of the calibrated radiance imagery with the SMARTS model[51] to derive surface reflectance spectra. We carried out ortho-rectification of the hyperspectral imagery (PARGE, ReSe Applications Schläpfer, Wil, Switzerland) with readings acquired by the inertial measuring unit on board the airborne platform (IG500 model, SBG Systems, France). We applied spatial binning through object-based image analysis, thus increasing the signal-to-noise ratio (SNR) of the instrument. Additionally, we conducted spectral binning to reduce the number of spectral bands (260 bands at 1.85 nm resolution). SNR reached >300:1 after binning. We acquired high-resolution tree-crown temperature images with a thermal camera (FLIR SC655, FLIR Systems, USA) at 640 × 480 pixels resolution using a 24.6 mm f/1.0 lens, sensitive to the 7.5–14 μm spectral range and sensitivity below 50 mK.

We identified individual trees in the high-resolution hyperspectral and thermal images using object-based crown detection and segmentation methods[52]. We then calculated the mean hyperspectral radiance, reflectance and temperature for each pure tree crown within every orchard under evaluation. We based our image segmentation methods on Niblack[53] and Sauvola and Pietikäinen[54], which allowed the isolation of tree crowns from the soil and shadow components. The segmentation of each tree crown was assessed visually to ensure a minimum number of pure vegetation pixels were selected within each tree crown and also spectrally to evaluate the purity of the reflectance extracted from the crown to avoid spectral mixture with soil, shadows and background components[24,35].

**Collection of *Xf* and *Vd* biotic stress field data**. Field assessments of *Xf*- and *Vd*-infected trees were carried out from outbreaks affecting olive and almond species in the indicated regions of Italy and Spain between 2011 and 2019[24,35,52]. During these campaigns, we performed quantitative PCR (qPCR)[55] for *Xf* in olive and almond (Alicante), recombinase-polymerase-amplification (RPA) using the AmplifyRP XRT + test (Agdia®, Inc., Elkhart, IN)[56] for *Xf* in almond (Majorca) or conventional PCR[57] assays for *Vd*, as well as visual assessments in individual trees of disease incidence (DI) and disease severity (DS). A sample was considered positive if Ct values were ≤36 and amplification curves were exponential. PCR/qPCR data for model analysis were transformed to 0 and 1, for negative and positive results, respectively, and Ct values were not used in the analysis (see Supplementary Table 2 for the PCR/qPCR primer sequences for *Vd* and *Xf*). DS was scored using a 0–4 rating scale according to the percentage of the tree crown showing disease symptoms.

In Apulia, the *Xf*-olive database comprised a total of 15 olive groves surveyed during the June 2016 and July 2017 campaigns. Visual assessments for infection were conducted on 7296 trees (3324 in 2016 and 3972 in 2017). In 2016, 1886 symptomatic (and 1438 asymptomatic) trees were surveyed (762 trees labelled as DS = 1; 802 DS = 2; 250 DS = 3 and 72 DS = 4). In 2017, 1365 were reported as symptomatic (and 2607 asymptomatic) (686 DS = 1; 542 DS = 2; 122 DS = 3 and 15 DS = 4). qPCR assays were carried out to diagnose *Xf* infection in 77 olive trees, whereby 39 trees tested negative (qPCR = 0) and 38 tested positive (qPCR = 1).

On the island of Majorca and at the Alicante province, the field-based *Xf*-almond database comprised a total of 19 almond groves surveyed in 2018 and 2019, respectively. In Alicante, the field surveys covered 83 ha with 9 almond groves consisting of 943 almond trees. During the field campaigns, almond trees were visually assessed to evaluate *Xf*-induced DI and DS indices. From this analysis, we identified 593 symptomatic and 350 asymptomatic trees. Out of all symptomatic trees, 163 were rated as DS = 1, 214 DS = 2, 157 DS = 3, and 59 DS = 4. Furthermore, qPCR analysis was carried out on 226 almond trees to diagnose *Xf* infection, resulting in 48 non-infected (qPCR = 0) almond trees and 178 infected trees (qPCR = 1). In Majorca, field surveys in July 2019 covered a total of 2803 ha and comprised 10 almond groves. During the field campaigns, visual observations were carried out on over 4048 almond trees to assess DI and DS, yielding 1387 symptomatic and 2661 asymptomatic trees. From symptomatic trees, 537 were rated as DS = 1 449 DS = 2, 359 DS = 3 and 42 DS = 4. We conducted AmplifyRP XRT + assays on 265 almond trees for diagnosing *Xf* infection the same day they were sampled and identified 141 negative trees (qPCR = 0) and 124 positive trees (qPCR = 1).

We carried out physiological measurements of leaf chlorophyll, anthocyanins, flavonoids and nitrogen contents with a Dualex Scientific + (Force-A, Orsay, France) instrument as well as leaf reflectance (400–1000 nm spectral range) and steady-state chlorophyll fluorescence (Ft) using the PolyPen RP400 and FluorPen FP100 instruments (Photon Systems Instruments, Drasov, Czech Republic) during the field evaluations of almond and olive groves conducted in Majorca, Alicante and Apulia regions. In the *Xf*-olive study site in Apulia, we generated 1023 leaf measurements with Dualex, 1543 single leaf reflectance spectra, as well as 1402 Ft readings over 67 olive trees. In the *Xf*-almond study sites in Majorca, we measured 1242 leaves with Dualex, 1094 leaves with the PolyPen and 1218 with the Fluorpen instruments from 87 almond trees across a wide range of disease severity levels. For the *Xf*-almond study sites located at Alicante, we conducted 1649 leaf measurements with Dualex, 632 leaf measurements with PolyPen and 563 leaf measurements with FluorPen FP100 over 43 almond trees.

We assessed *Vd*-infected olive trees from 11 olive groves by surveying an area of over 3000 ha in Castro del Rio and Écija, southern Spain, in 2011 and 2013, respectively. In Castro del Rio, we conducted visual assessments in an infected area of 96 ha comprising 1878 olive trees, thus identifying 1569 asymptomatic and 283 symptomatic olive trees. Out of the 283 symptomatic trees, 218 were rated as DS = 1; 45 DS = 2; 12 DS = 3 and 8 DS = 4. We measured leaf Fs and Fm' fluorescence parameters from 25 leaves per tree using a PAM-2100 Pulse-Amplitude Modulated Fluorometer (Heinz Walz GMBH, Effeltrich, Germany). In addition, leaf PRI$_{570}$ was measured from 25 leaves per tree using a custom-made PlantPen device (Photon System Instrument, Drasov, Czech Republic). Finally, we measured leaf conductance (Gs) on five leaves per tree using a leaf porometer (model SC-1, Decagon Devices, Washington, DC, USA). In the Écija region, the surveyed area covered 3424 ha, and 5223 olive trees were evaluated. We performed visual assessment to determine DI and DS indices of *Vd*-infected trees, identifying 5040 asymptomatic olive trees. Of the remaining 183 olive trees that were symptomatic, 112 were trees rated as DS = 1; 41 DS = 2; 22 DS = 3 and 8 DS = 4.

Trees were evaluated for disease severity and incidence by visual assessment in each outbreak region. PCR assays were carried out on a subset of these trees within each orchard to (i) validate that the pathogen (*Xf* or *Vd*) was actually present and the biotic source of symptoms; and (ii) validate that asymptomatic (DS = 0) but infected (PCR = 1) trees were detected using the hyperspectral plant traits estimated through the methodology described in this paper. In general, PCR assays are (i) time consuming and costly, and (ii) difficult to make in large infected trees due to the non-uniform distribution of the infection within each tree crown. These PCR data for each tree along with the field evaluations of DS, DI and non-destructive physiological measurements derived for each tree within every orchard were matched with the high-resolution hyperspectral images to build the biotic databases used in this study. We carried out the field work at each orchard guiding the evaluations and measurements using a high-resolution image to map the location of each tree within the orchard. Due to the planting grids typical of almond and olive species, which were not contiguous or in row-structured patterns, the identification of each individual tree in the images was straightforward.

**Collection of abiotic stress field data**. We monitored over 3600 ha of olive and almond groves located outside any infected area in Cordoba and Seville, Southern Spain. We performed a multitemporal analysis to study the spectral plant-trait alterations induced by abiotic stress relative to healthy olive and almond trees with data we collected over a 468 ha area comprising two olive and one almond groves throughout July 2016 and August 2017 growing seasons. We analysed 2975 olive and 1964 almond trees in 2016, and 2865 olive and 2063 and almond trees in 2017. At both study sites, we monitored the midday stem water potential (SWP) using a pressure chamber (Soil Moisture Equipment Corp. model 3000, Santa Barbara, CA, USA) on 16 trees per grove. SWP values showed differences between two existing irrigation levels (well-watered and mild water stress), averaging −1.7 and −1.9 MPa across the season in the case of almonds. In olive, SWP in one of the groves reached −3.8 and −3.5 MPa. In 2017, water potential levels averaged −2.9 and −2.0 MPa. In the second grove, irrigation levels were higher, reaching an average SWP of −1.5 MPa. We used an additional study site located in Casariche (Seville province), southern Spain, to validate the results obtained from the multitemporal analysis. This study site covered 3371 ha containing 55 olive groves grown under irrigated and rainfed conditions, with 21,071 olive trees used for statistical analysis.

The multitemporal dataset was used to evaluate the water-induced abiotic stress by quantifying the evolution of the importance of the most sensitive spectral traits by clustering non-stressed trees ($C_0$) against groups of trees exposed to increasing levels of water stress ($C_1$ to $C_4$). The multitemporal component of this assessment enabled the evaluation of every single tree across time, therefore selecting the trees for each cluster based on a sustained water stress level, avoiding the selection of trees under short-term stress dynamics. Thus, the clusters were determined based on their CWSI levels, and only the trees with stable water stress levels across two consecutive years (2016 and 2017) were selected for the analysis. For this purpose, we did not include trees that deviated beyond 95% of the CWSI differences calculated between the first and second year in the analysis. After this trimming step, we retained 5484 olive trees (from 5566 trees) and 3652 almond trees (from 3882 almond trees). Trees were then grouped through CWSI clustering analysis using a modified three-sigma rule[58]. This rule describes the density of a distribution within standard deviation bands on both sides of the mean point into the 68th, 95th and 99.7th percentiles[58], representing μ ± σ, μ ± 2σ and μ ± 3σ, respectively. The first interval defined by the classic three-sigma rule (μ ± σ) represented most trees, while the third interval (μ ± 3σ) consisted of very few trees, raising issues for the determination of statistical significance analysis. Based on this observation, we adjusted the breakpoints between groups as follows: we classified those trees that were in the lowest 10th percentile as $C_0$. Trees between the 10th and 68th percentiles (μ + σ) were classified as $C_1$, trees between the 68th and 85th percentile were classified as $C_2$, trees between the 85th and 95th percentile were classified as $C_3$ and finally the trees over the 95th (μ + 2σ) percentile were classified as $C_4$. We thus selected 488 $C_0$, 3066 $C_1$, 1090 $C_2$, 618 $C_3$ and 222 $C_4$ olives trees. Likewise, we grouped almond trees into 390 $C_0$, 1776 $C_1$, 1248 $C_2$, 214 $C_3$ and 24 $C_4$ clusters. Moreover, the analysis of the contribution of a given trait was performed using ML modelling strategies to classify unstressed trees against the clusters defined above that were exposed to increasing levels of water stress. Furthermore, we assessed the consistency of the obtained indicators by performing the classification between stressed and non-stressed trees at an independent olive study site. For this purpose, we evaluated our predictors and compared their contribution over an additional site (Casariche).

**Model inversion methods for plant-trait estimation**. We quantified chlorophyll content ($C_{a+b}$), carotenoid content ($C_{x+c}$), anthocyanin content (Anth.), mesophyll structure (N), leaf area index (LAI) and average leaf angle (leaf inclination distribution function or LIDF) by radiative transfer model inversion of PROSPECT-D[59] and 4SAIL[60], as in Zarco-Tejada et al.[24]. We inverted PROSPECT-D + 4SAIL using a look-up-table (LUT) generated with randomised input parameters. The LUT was generated with 100,000 simulations within fixed ranges (Supplementary Table 3). We implemented a wavelet analysis[61] into six wavelets by a Gaussian kernel, estimating the parameters in the top 1% entries ranking the lowest root mean square error (RMSE) values. We then retrieved each plant trait independently by training supported vector machine (SVM) algorithms using the simulated reflectance data as input. We built SVMs in Matlab (MATLAB; Statistics and Machine Learning toolbox and Deep Learning toolbox; Mathworks Inc., Matick, MA, USA) using a Gaussian kernel (radial basis function) with hyperparameters optimised for each model. The training processes were carried out in parallel using the Matlab parallel computing toolbox. With these trained models, we then used the spectral reflectance extracted from the delineated crowns (as show in Fig. 1) to predict plant traits for each individual tree at each study site. The model inversions were carried out for each tree using the crown reflectance. The latter was calculated as an average across all the pixels belonging to the tree crown, delineated using segmentation. This method[52] avoids the problem of pixels from within-crown shadows, from tree edges or from sunlit or shaded soil background affecting the spectra, as it retrieves the plant traits from pure sunlit vegetation components of the trees. We also calculated narrow-band spectral indices from reflectance spectra (Supplementary Table 1), which are sensitive to leaf traits and potentially related to disease-induced symptoms. Tree-crown radiance and temperature were used to calculate sun-induced chlorophyll fluorescence at 760 nm (SIF$_{@760}$) and the crop

water stress index (CWSI)[37]. SIF$_{@760}$ was quantified using the O$_2$-A *in-filling* Fraunhofer Line Depth (FLD) method[63] and CWSI was calculated by incorporating the tree temperature and the weather data obtained at each study site[37].

**Statistical analysis**. We implemented random forest (RF)[64] algorithms to classify healthy vs. infected (biotically stressed) trees, and non-stressed vs. water (i.e. abiotically) stressed trees for both tree species. RF algorithms have been widely used in remote sensing studies since they have shown excellent classification accuracies and high processing speeds with high-dimensional data[62] and have shown to be accurate in detection of several diseases[29,65–67]. The spectral plant traits estimated by radiative transfer model inversion (C$_{a+b}$, C$_{x+c}$, Anth., LAI and LIDF), CWSI and SIF$_{@760}$ were used as inputs for the models. In addition, using a recursive feature elimination approach[68] the narrow-band indices that improved the classification in terms of overall accuracy (OA) and kappa coefficient (κ) were added to the models. The pool of narrow-band indices was reduced based on a variance inflation factor (VIF) analysis[69] to avoid collinearity among the input features.

The RF algorithms were built in Matlab and the hyperparameters were optimised using Bayesian optimisation. The importance of a feature using the RF algorithm was assessed based on the permutation of out-of-bag (OOB) predictor methodology[70]. To compare the relative differences of the spectral traits in classification of the biotic and abiotic stress, the importance was normalised by dividing the importance of each trait by the highest contribution obtained for each pathogen/species. For the RF models, 500 iterations were run by randomly partitioning each dataset into training (80% of samples) and testing sets (20% of samples). For the training subset, a balanced number of samples from each class was randomly selected at each iteration. The importance obtained by the OOB permutation algorithms was used to build a feature-weighted random forest algorithm (based on Liu and Zhao[45]), accounting for the importance of each variable on the classification process, evaluating the model against PCR data and visual observations for each biotic stress dataset in terms of OA and κ levels.

Probabilities of the predictions were obtained for each sample[71] and the uncertain trees were assessed. To extract the uncertainty for each individual tree on the classification, we evaluated the probability distribution for each class from each dataset independently. Then, those trees with a classification probability below the 68$^{th}$ percentile (μ [mean] + σ [standard deviation]) were considered as uncertain and incorporated into a second-stage classification process. The second stage consisted of an unsupervised graph theory–based spectral clustering algorithm[72] and included traits selected by focusing on the divergent biotic–abiotic stress obtained from the biotic and the abiotic stress databases. Spectral clustering was performed in R using the kernlab package[73].

To determine the spectral traits that differed between *Xf*- and *Vd*-infected plants and those from the abiotic pathway, we first normalised the importance of the specific traits independently. Then, we compared the common traits between abiotic and biotic stress sets, selecting only biotic stress-related traits that differed in ratio by >0.5 over their homologous abiotic stress trait values. Traits that were only expressed under biotic stress conditions and that showed a normalised importance over 0.5 were included for the second-stage classification process only including those divergent-specific biotic and abiotic stress-related spectral traits as inputs. Specifically, NPQI, Anth. and SIF$_{@760}$ were considered for the classification of *Xf*-infected olive trees. C$_{a+b}$, SIF$_{@760}$ and PRI$_n$ were used for classifying *Xf*-infected almond trees. Furthermore, NPQI, Anth. and B spectral traits were selected for classifying uncertain *Vd*-infected olive trees. Finally, we validated our feature-weighted methodology coupled with the second-stage spectral clustering method against qPCR assays and visual assessment of symptom severity.

**Reporting summary**. Further information on research design is available in the Nature Research Reporting Summary linked to this article.

## Data availability

The data used in this study[74] are available at the repository https://github.com/HyperSens/HyperSens-Divergent-spectral-responses-Nature-Communications and can be cited as https://doi.org/10.5281/zenodo.5535095. Due to the large size of the airborne hyperspectral datasets used in this study, the image shown in Fig. 1 is available upon reasonable request contacting P.J.Z.-T.

## Code availability

The codes used for this study[74] are available at the repository https://github.com/HyperSens/HyperSens-Divergent-spectral-responses-Nature-Communications and can be cited as https://doi.org/10.5281/zenodo.5535095. Additional algorithms required for specific intermediate steps are available upon reasonable request contacting P.J.Z.-T.

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

## Acknowledgements

We thank QuantaLab-IAS-CSIC staff members A. Vera, D. Notario and R. Romero for laboratory assistance, and M. Morelli, L. Susca, M. Montes-Borrego, G. Leon, J.L. Trapero-Casas and D. Sacristán for their strong support during the field campaigns. We are also grateful for the information provided by the Plant Health Service of the General Directorate of Agriculture of the region of Valencia (Spain) and the Agricultural Service of the General Directorate of Agriculture, Livestock and Rural development of the Balearic Islands (Spain) on the phytosanitary status of almond orchards in the affected area of Alicante and Majorca, respectively, and to TRAGSATEC (Grupo TRAGSA) for their support during the field campaign in Alicante. The study was partially funded by the European Union's Horizon 2020 Research and Innovation Programme through grant agreements POnTE (635646) and XF-ACTORS (727987), as well as by projects AGL2009-13105 from the Spanish Ministry of Education and Science, P08-AGR-03528 from the Regional Government of Andalusia and the European Social Fund, project E-RTA2017-00004-02 from 'Programa Estatal de I + D + I Orientada a los Retos de la Sociedad' of Spain and FEDER, Intramural Project 201840E111 from CSIC, and Project ITS2017-095 Consejeria de Medio Ambiente, Agricultura y Pesca de las Islas Baleares, Spain. The views expressed are purely those of the writers and may not in any circumstance be regarded as stating an official position of the European Commission.

## Author contributions

P.J.Z.-T. and T.P. designed the objectives of this study; P.J.Z.-T., T.P., V.G.-D., P.S.A.B., B.B.L., D.B., M.S. and J.A.N.-C. designed research; P.J.Z.-T., T.P., C.C., V.G.-D., R.C., A.H., R.H.-C., M.R.-E., M.P.V.-A., B.B.L., P.S.A.B., M.S., D.B. and J.A.N.-C. performed research; P.J.Z.-T., T.P., C.C., V.G.-D., R.C., A.H., R.H.-C., M.R.-E., M.P.V.-A., B.B.L., P.S.A.B. and J.A.N.-C. analysed data; P.J.Z.-T. and T.P. wrote the paper, and all authors contributed and provided comments, read and approved the final submission.

## Competing interests

The authors declare no competing interests.
