## [Peer Review File · Nature Communications]

Divergent abiotic spectral pathways unravel pathogen stress signals across speciesREVIEWER COMMENTS

Reviewer #1 (Remarks to the Author):

The manuscript describes a highly relevant research topic on the usability of combined spectral remote sensing observations to detect biotic stress in cropping systems. And in addition, present an approach to disentangle biotic stress responses from abiotic effects. Although more research has been done on this topic, this is mostly limited to one or a number of fields and/or for one type of disease. The dataset presented in this manuscript is extensive and allows a more general synthesis. The experimental set-up is clear but requires some more detailed explanation at some points as described below. The figures support the analysis results, at some points some changes need to be made to achieve better understanding for the readers. One final remark would be relevant to add in relation to the starting point of the study, in what way could the presented approach be adopted to investigate other diseases. What would be required and how disease/crop specific is the presented approach.

P2 par1: here a list is provided of threatening pathogens: would be more relevant when also a reference could be provided with a link to remote sensing based opportunities for detection of this disease. Then the list would become more relevant within the scope of the manuscript

P5 Figure 1: what is the actual method which provides the values for the normalised importance. This is not specifically described in the methods section and would be good to add in the caption of figure 1 as other statistical methods are explicitly mentioned here. Its not clear what is the source of the purple boxes NPQI & B traits: would be clear to refer to methods table. The title of figure 1 (healthy vs. infected) is not self-explanatory. Which bars are healthy and which infected?

P6 paragraph 2: the validation results which are presented here, do they include the whole range of sampling conditions so including trees with abiotic stress?

P7 paragraph 1: focus here is on the analysis of temporal data to detect abiotic stress factors: its not clear why a temporal dataset would be required, could this also be achieved with a one-moment observation with varying moisture/water stress conditions within the selection of orchards?

P8 Figure 2: based on the chosen colors, no distinction can be made for the purple and brown bars. It seems that for stage S4, a purple bar received a blue color.

P10: Figure 3: the acronym L and E in panel f need explanation.

P13 Methods par2: can you provide information on the accuracy of the segmentation result and how the location of the segmented trees was matched with the field observations. Trees in the field were located using RTK or did you apply other approaches? How sure can you be that you match the right tree in the image with the information at tree level observed in the field for example in relation to achieved geometrical accuracy

P13 Field data collection: In paragraph 2 of this section you describe the field observation for Xf and the PCR assessment: from the text its not clear how will the field observations represented the PCR test result.

Idem for next paragraph.

P14 paragraph Abiotic stress dataset: here its mentioned that a multi-temporal analysis was executed: can you provide details on reason why this was executed on more time periods and how the comparison of different time-points was made in relation to the disease level of the trees.

P15 For which scale level did you prepare the RTM inversion: was this at the pixel level or for the RS signal of the segmented trees. Would this make a difference in the retrieved plant traits as used in this research?

P15 Statistical analysis: the choice was made to combine/compare the RF and SVM method: from description it's not apparent how the modelling results of both methods were used for the uncertainty assessment using the 68th percentile as threshold. The procedure for the second-stage classification process needs additional explanation: which training procedure was used here?

P20 Table 2: there seems to be disbalance between the large number of spectral indices and the SIF and thermal derived indices. When these are used as input for the ML methods, will these methods be able to highlight the importance of the individual thermal index in relation to the spectral indices.

*** final point: next time add line numbers to the mauscript ***

Reviewer #2 (Remarks to the Author):

In this study, conventional machine learning algorithms including RF and SVM were developed to classify healthy vs. infected (biotic stress) trees, and water stressed vs. non-water stressed (abiotic stress) trees in both tree species. However, these textbook level methods are not novel, which makes the paper difficult to be published in such a high-quality journal.

Reviewer #3 (Remarks to the Author):

Authors analyzed over one million olive and almond trees using an airborne imaging spectroscopy and thermal imaging facility targeting infected and healthy trees. The biotic and abiotic stress datasets are valuable to the society. Authors used a feature-weighted ML algorithm to unravel these confounding spectral alterations induced by biotic and abiotic stressors. The work is novel in identifying biotic and abiotic stresses using big data and machine learning approaches but may not be of interest to broader community.

I have several major concerns regarding the design of the experiments:

1. Authors used olive and almond trees as examples but these types of trees are used for different purposes. This is not easy for readers to follow and provides no significant contribution for comparison as illustrated in Figure 1. The spectral changes of olive trees are used to identify divergent Xf- and Vd-induced biotic-abiotic spectral alterations while the spectral changes of almond trees are used to identify the drivers from abiotic stress. Figure 1 is somewhat confusing with Figure 3. I would suggest to focus Xf-induced biotic-abiotic spectral alterations for both olive and almond and further analyze the conditions of irrigated or not for both trees as well.
2. The manuscript lacks clear flow of the information and results. Authors should provide an improved design of experiments to illustrate the common and unique findings to unravel confounding spectral alternations induced by biotic and abiotic stressors.
3. Detailed data illustration for each experiment should be provided to better understand the group of samples. Authors should provide further analyses and discussions on whether different areas and years would impose additional divergent pathways.
4. Authors claimed “hyperspectral methods advance early detection of devastating pathogens” in the abstract. The current manuscript, however, does not provide sufficient support on in-season detection and lacks on discussion of different contributions by hyperspectral and thermal images. Authors should provide further analyses on divergent contributions of hyperspectral and thermal images.

Minor comments:

1. Citations should be removed in the abstract.
2. The limitations and further implications of the study are not well written.

Answers to the Reviewers' comments

We would like to thank all reviewers for their comments and suggested improvements. We have indicated our answers in red.

Reviewer #1

The manuscript describes a highly relevant research topic on the usability of combined spectral remote sensing observations to detect biotic stress in cropping systems. And in addition, present an approach to disentangle biotic stress responses from abiotic effects. Although more research has been done on this topic, this is mostly limited to one or a number of fields and/or for one type of disease. The dataset presented in this manuscript is extensive and allows a more general synthesis. The experimental set-up is clear but requires some more detailed explanation at some points as described below. The figures support the analysis results, at some points some changes need to be made to achieve better understanding for the readers. One final remark would be relevant to add in relation to the starting point of the study, in what way could the presented approach be adopted to investigate other diseases. What would be required and how disease/crop specific is the presented approach.

Thank you for the review and comments formulated, which helped improve the manuscript and make it clearer. We have tried to answer all questions and revised the paper accordingly.

As suggested, we have made changes in the figures and throughout the paper, and added a final remark indicating how to apply our findings to other diseases and how disease/crop specific this approach is. We also added a sentence on the future of hyperspectral imagers onboard satellites for global monitoring of diseases (pages 11-12, lines 309-340):

"The work presented here demonstrates that potentially confounding symptoms of biotic and abiotic stress can be distinguished for particular host plant species. Our analyses of the most comprehensive high-resolution imaging dataset of pathogen-specific hyperspectral traits compiled so far show, for the first time, the existence of host- and pathogen-specific spectral plant responses that diverge between biotic and abiotic stresses. Our work goes beyond current knowledge, accurately detecting harmful xylem-limited pathogens across host species.

Global warming and international trade are exacerbating risks related to emerging and reemerging pathogens threatening agriculture. At the same time, world food production needs to increase by 50% over the next 30 years to feed a growing global population, despite decreasing arable land and climate disruption⁴⁸. With yield losses due to pathogens exceeding 30% in regions where food security is critical, the development of technologies for large-scale early detection of outbreaks is crucial. A global plant disease monitoring framework will require collaboration across disciplines, including remote sensing, physics, artificial intelligence, engineering and sensor development, and space and drone industries, interacting closely with plant pathology, physiology, and agronomy.

The analytical approach introduced here provides a transferrable framework to disentangle pathogen-induced stress from abiotic dynamics in a range of species, which is critical for the development of global disease-detection models. Detecting the coexistence of both factors is also fundamental to evaluate the evolution and treatment of the plant, either to adapt the treatment in case there is only one stress factor, or to control the interaction of biotic and abiotic

stresses. For example, drought can play a key role in the development of plant diseases^{49,50}. The applicability of this framework to other pathotypes will require further considerations; for individual plant diseases, it will depend on the degree of divergence of the spectral pathways induced by the coupled biotic–abiotic stress-related physiological alterations for each species. We expect results to further improve for non-xylem-limited pathogens that cause physiological responses uncoupled from the abiotic dynamics of water stress. Widespread use will require further developments of technology readiness; a critical limitation for the operational application of these methods lies in the need for high-spatial resolution hyperspectral and thermal imaging (i.e., at sub-meter resolutions), a technology currently available only from drones at small scale, and from manned airborne platforms such as the ones used here. Future hyperspectral sensors on board satellites or high-altitude drones may enable systematic data collections with imaging spectroscopy at sub-meter resolution data, and, when combined with analytical frameworks, permit the real-time monitoring of diseases and abiotic stresses at global scales.”

P2 par1: here a list is provided of threatening pathogens: would be more relevant when also a reference could be provided with a link to remote sensing based opportunities for detection of this disease. Then the list would become more relevant within the scope of the manuscript

Thank you for the suggestion. In such a paragraph, remote sensing technology hasn't been introduced yet; therefore, citing the published work on remote sensing detection methods might be confusing. As suggested, we have added more remote sensing – plant diseases references instead in a paragraph on the same page, after the need for remote sensing for the early detection of diseases is discussed (page 2, lines 72-75).

*“The development of robust large-scale plant scanning methods will be key to successfully monitor detrimental crop pathogens and assist in their timely eradication or optimise containment measures²³. Advanced imaging spectroscopy is the only large-scale method that allows early detection of infectious plant diseases, i.e. when symptoms are not visible yet but spread of the pathogen can occur²⁴. **Hyperspectral imaging has been recently used to detect, for example, rice sheath blight²⁵, tobacco mosaic virus²⁶, late blight, target and bacterial spots²⁷, spotted wilt virus in tomato²⁸, phytophthora-induced decline²⁹, verticillium wilt and the olive quick decline syndrome³⁰.** However, a major limitation of advanced hyperspectral, thermal scanning and radiative transfer methods in plant health monitoring is that the subtle physiological alterations caused by a disease reflect changes in plant physiological state, such as stomatal regulation³¹ and the coupled chlorophyll fluorescence-photosynthesis-transpiration dynamics³², which are all commonly modulated by both biotic and abiotic confounding factors. Revealing distinct spectral fingerprints associated with biotic- vs. abiotic-stress conditions is thus of paramount importance for large-scale remote detection efforts of early disease infection symptoms that occur in the context of natural physiological variability (e.g., due to water deficit or nutrient deficiencies) commonly found even in irrigated croplands.”*

P5 Figure 1: what is the actual method which provides the values for the normalised importance. This is not specifically described in the methods section and would be good to add in the caption of figure 1 as other statistical methods are explicitly mentioned here. Its not clear what is the source of the purple boxes NPQI & B traits: would be clear to refer to methods table. The title of figure 1 (healthy vs. infected) is not self-explanatory. Which bars are healthy and which infected?

Thanks for raising this question about the normalised importance, which was probably not well described. As suggested, we have added a description regarding the normalised importance in the Main Text, Figure 1 caption (now Figure 2), and in Method section, and also referred to the Extended Data Table 2 in the Figure 1 (now Figure 2) caption:

Page 5, Lines 136-140.

“To make the results comparable across species and pathogens, the obtained importance for each spectral trait was normalised by the highest importance of each pathogen/species within each ML model (see Methods for detailed description). This approach revealed, on a common scale, how important individual spectral plant traits were in the overall response of the two host species studied here to biotic and abiotic stressors.”

Figure 1 (new Figure 2) caption:

“Fig. 2 | Importance of spectral traits to detect Xf- and Vd-infection symptoms. a–d Normalised importance of hyperspectral and thermal plant traits retrieved from the pool of spectral plant traits used to detect Verticillium dahliae, Vd- (a) and Xylella fastidiosa, Xf-induced infection symptoms (b,c,d) across olive (a,b) and almond (c,d) trees. For reference, the full list of spectral plant traits is available in the Extended Data, Table 2. The importance analysis was carried out using a balanced training dataset obtained from n=1,878 (a), n=7,296 (b), n=4,048 (c), n=2,680 (d) trees by the permutation of out-of-bag (OOB) predictor methodology. The importance of each spectral trait was normalised by the highest importance obtained for each disease/species within each ML model. e–j Analysis of spectral plant traits measured in the field from asymptomatic vs. Xf-infected olive and almond leaves. e, Temperature at midday (t, n=2,584 leaf samples). f, Normalised phaeophytinization index-based Spectral Trait (NPQI, n=1,457 leaf spectral samples). g, Anthocyanins (Anth, n=1,318 leaf samples). h, Steady-state leaf chlorophyll fluorescence (Ft, n=2,887 leaf samples). i, Normalised xanthophyll cycle dynamics index (Photochemical Reflectance Index [PRI_n], n=1,457 leaf spectra samples). j, leaf chlorophyll content (C_{a+b}, n= 2,584 leaf samples). Statistical analyses were carried out by a Kruskal-Wallis test followed by a Wilcoxon post-hoc test with Bonferroni correction to examine significant differences at p < 0.05 between the leaf groups for each species. Severity levels with the same letter are not significantly different (p-value ≥0.05). The horizontal black line in the boxplots displays the median, and the top and bottom horizontal lines represent the 75th and 25th percentiles, respectively. Whiskers display the lower and upper limits of interquartile ranges (Q±1.5xIQR).”

Methods section (page 20, lines 660-665):

*“The importance of a feature using the RF algorithm was assessed based on the permutation of out-of-bag (OOB) predictor methodology⁷⁰. **To compare the relative differences of the spectral traits in classification of the biotic and abiotic stress, the importance was normalised by dividing the importance of each trait by the highest contribution obtained for each pathogen/species.** For the RF models, 500 iterations were run by randomly partitioning each dataset into training (80% of samples) and testing sets (20% of samples).”*

We have also added titles to all figures to make them self-explanatory and clarified that the bars show the importance of spectral traits to disease detection.

P6 paragraph 2: the validation results which are presented here, do they include the whole range of sampling conditions so including trees with abiotic stress?

Thanks for the question, which shows that the text was not sufficiently clear. In the results mentioned in such paragraph (page 8, paragraph starting with line 200), the variability of the abiotic-induced stress was not yet accounted for in the models used to detect both pathogen-induced symptoms (X_f , V_d). The objective was to highlight the high uncertainty in biotic detection when abiotic stress is not accounted for. Because of this, the model performance reported in such a paragraph was lower than the ones reported later when the full methodology accounting for the abiotic-stress detection was adopted. We have clarified this aspect in the main text, indicating that the results shown in such a paragraph were obtained without taking into account abiotic-induced variability. We added the following sentence (in bold) to clarify this aspect (page 8, lines 209-212):

*“Our feature-weighted models, **which at this stage account only for the biotic stress-induced variability in the spectral traits** yielded OA=84% ($\kappa=0.68$) for X_f detection of infection in almond ($n=4,048$ trees), and OA=77% ($\kappa=0.43$) and OA=75% ($\kappa=0.49$) for X_f and V_d detection of infection in olive, respectively ($n=7,296$ and $n=1,852$ trees, respectively).”*

The paragraph below (lines 224-234) then introduces the need for accounting for the abiotic variability for improved detection results:

*“We disentangled the changes in spectral plant traits caused by biotic (X_f and V_d infection) and abiotic stressors by analysing a temporal series of airborne imaging spectroscopy and thermal imagery acquired in areas free of pathogens yet experiencing variable water status levels. We flew over ca. 380,000 olive and almond trees across three geographical regions with our airborne imaging sensors over two summer growing seasons. **This multitemporal dataset enabled the identification of individual trees that showed consistent and sustained water status levels across seasons. The temporal approach allowed clustering of trees with different stress levels over a long period of time (i.e., two growing seasons) rather than focusing the analysis on short-term water stress conditions potentially due to transitory environmental effects or irrigation system malfunctions in each orchard under study. Thus, the multiyear dataset improved the selection of trees consistently experiencing sustained water stress**”.*

P7 paragraph 1: focus here is on the analysis of temporal data to detect abiotic stress factors: its not clear why a temporal dataset would be required, could this also be achieved with a one-moment observation with varying moisture/water stress conditions within the selection of orchards?

Thanks for this question, which also shows that we were not sufficiently clear in explaining the reason for using the multi-temporal abiotic dataset. We have indicated in the Main text the reason for using the temporal dataset instead of a one-moment snapshot of the water stress status (page 9, lines 228-234) as indicated above.

P8 Figure 2: based on the chosen colors, no distinction can be made for the purple and brown bars. It seems that for stage S4, a purple bar received a blue color.

Thank you for noticing the error in one of the bars. This issue has been solved, and the use of different colours for the purple and brown bars. A title has also been added to make the figure self-explanatory.

P10: Figure 3: the acronym L and E in panel f need explanation.

Thank you, change made in Figure 3 (now Figure 4).

P13 Methods par2: can you provide information on the accuracy of the segmentation result and how the location of the segmented trees was matched with the field observations. Trees in the field were located using RTK or did you apply other approaches? How sure can you be that you match the right tree in the image with the information at tree level observed in the field for example in relation to achieved geometrical accuracy

Thank you for this question. We agree that we have not described it properly. We have clarified this issue in the Methods in two sections:

Page 16. Lines 502-505.

“The segmentation of each tree crown was assessed visually to ensure a minimum number of pure vegetation pixels were selected within each tree crown and also spectrally to evaluate the purity of the reflectance extracted from the crown to avoid spectral mixture with soil, shadows and background components^{24,35}”.

Page 18. Lines 568-575.

“These PCR data for each tree along with the field evaluations of DS, DI, and non-destructive physiological measurements derived for each tree within every orchard were matched with the high-resolution hyperspectral images to build the biotic databases used in this study. We carried out the field work at each orchard guiding the evaluations and measurements using a high-resolution image to map the location of each tree within the orchard. Due to the planting grids typical of almond and olive species, which were not contiguous or in row-structured patterns, the identification of each individual tree in the images was straightforward.”

In addition, we felt that the paper required a new Figure to clearly describe the type of hyperspectral datasets used in this study, illustrating the details about the spatial and spectral resolution of the images. Based on the questions formulated by the reviewer, we think that the new Figure 1 clarifies aspects about the location of the trees, identification of the pure vegetation pixels, and the general features of the spatial and spectral image resolution. Below is a small capture of the new Figure 1 (Page 4):

106 **Fig. 1 | High-resolution airborne hyperspectral image acquired over one of the Apulia *Xylella***
 107 ***fastidiosa* (*Xf*)-infected areas. Similar datasets were collected from all *Xf* and *Verticillium dahliae***
 108 **(*Vd*) outbreaks used for the analyses carried out in this study. a, mosaic covers 1450 ha at 40 cm**
 109 **resolution collected in the 400-885 nm range with 260 spectral bands. b, individual trees could be**
 110 **identified on the images and properly located during field work. Tree-crown segmentation**
 111 **algorithms (c) were used for the selection of pure vegetation pixels. Extracted reflectance (d) and**
 112 **radiance spectra (e) were used to calculate spectral indices, plant traits by radiative transfer model**
 113 **inversion, and solar-induced fluorescence (SIF) used as inputs for the disease detection models.**

4

P13 Field data collection: In paragraph 2 of this section you describe the field observation for Xf and the PCR assessment: from the text its not clear how will the field observations represented the PCR test result.

Idem for next paragraph.

A large number of trees were evaluated by visual assessment, a standard method used by plant pathologists to assess the disease incidence (DI, i.e., presence/absence of disease symptoms) and disease severity (DS, using 0 to 5 rating scale according to the percentage of the tree crown showing disease symptoms) of individual plants and trees. This method allowed the evaluation of thousands of trees within each outbreak. In addition, PCR assays were carried out in a limited number of leaf samples per orchard to i) demonstrate that the pathogen was actually present (*Xf* and *Vd*); and ii) to use these

trees with PCR data to demonstrate that asymptomatic (DS=0) but infected (PCR=1) trees were detected using the hyperspectral plant traits produced through the methodology described in this paper. It is not possible to carry out PCR assays for all trees evaluated because i) it is time-consuming and costly, and ii) the selection of leaf samples in large infected trees is difficult due to the non-uniform distribution of the infection within each tree crown, making sampling leaves from large tree crowns a challenge. To the best of our knowledge, the PCR data used in this study is the largest available worldwide in the context of *Xf* and *Vd* infected trees, and the only one with matched hyperspectral and thermal datasets.

We have clarified this aspect in the Methods section, as suggested:

Page 18, lines 562-575:

*“Trees were evaluated for disease severity and incidence by visual assessment in each outbreak region. PCR assays were carried out on a subset of these trees within each orchard to i) validate that the pathogen (*Xf* or *Vd*) was actually present and the biotic source of symptoms; and ii) validate that asymptomatic (DS=0) but infected (PCR=1) trees were detected using the hyperspectral plant traits estimated through the methodology described in this paper. In general, PCR assays are i) time consuming and costly, and ii) difficult to make in large infected trees due to the non-uniform distribution of the infection within each tree crown. These PCR data for each tree along with the field evaluations of DS, DI, and non-destructive physiological measurements derived for each tree within every orchard were matched with the high-resolution hyperspectral images to build the biotic databases used in this study. We carried out the field work at each orchard guiding the evaluations and measurements using a high-resolution image to map the location of each tree within the orchard. Due to the planting grids typical of almond and olive species, which were not contiguous or in row-structured patterns, the identification of each individual tree in the images was straightforward.”*

P14 paragraph Abiotic stress dataset: here its mentioned that a multi-temporal analysis was executed: can you provide details on reason why this was executed on more time periods and how the comparison of different time-points was made in relation to the disease level of the trees.

Thanks for this question, which helped to make the paper clearer. As indicated above, we have added the following paragraph in the Main Text:

Page 9, lines 224-234:

*“We disentangled the changes in spectral plant traits caused by biotic (*Xf* and *Vd* infection) and abiotic stressors by analysing a temporal series of airborne imaging spectroscopy and thermal imagery acquired in areas free of pathogens yet experiencing variable water status levels. We flew over ca. 380,000 olive and almond trees across three geographical regions with our airborne imaging sensors over two summer growing seasons. **This multitemporal dataset enabled the identification of individual trees that showed consistent and sustained water status levels across seasons. The temporal approach allowed clustering of trees with different stress levels over a long period of time (i.e., two growing seasons) rather than focusing the analysis on short-term water stress conditions potentially due to transitory environmental effects or irrigation system malfunctions in each orchard under study. Thus, the multiyear dataset improved the selection of trees consistently experiencing sustained water stress**”.*

In the Methods section, we have also clarified this aspect, moving one of the paragraphs to improve the flow, and modified the text to explain the multitemporal component of the abiotic assessment:

Page 18, lines 592-599:

“The multitemporal dataset was used to evaluate the water-induced abiotic stress by quantifying the evolution of the importance of the most sensitive spectral traits by clustering non-stressed trees (C0) against groups of trees exposed to increasing levels of water stress (C1 to C4). The multitemporal component of this assessment enabled the evaluation of every single tree across time, therefore selecting the trees for each cluster based on a sustained water stress level, avoiding the selection of trees under short-term stress dynamics. Thus, the clusters were determined based on their CWSI levels, and only the trees with stable water stress levels across two consecutive years (2016 and 2017) were selected for the analysis. For this purpose, we did not include trees that deviated beyond 95% of the CWSI differences calculated between the first and second year in the analysis.”

P15 For which scale level did you prepare the RTM inversion: was this at the pixel level or for the RS signal of the segmented trees. Would this make a difference in the retrieved plant traits as used in this research?

Thank you for this question. Due to the high-resolution imagery used, several pixels could be extracted from each tree crown. Nevertheless, we used the mean tree crown reflectance to estimate the plant traits by inversion, rather than using single pixels. This is a standard methodology used in several previous studies to estimate parameters representing the entire tree crown, avoiding single pixels from non-vegetation components. This aspect has been clarified in the text and also in the new Figure 1:

Page 19, lines 634-640:

“With these trained models, we then used the spectral reflectance extracted from the delineated crowns (as shown in Figure 1) to predict plant traits for each individual tree at each study site. The model inversions were carried out for each tree using the crown reflectance. The latter was calculated as an average across all the pixels belonging to the tree crown, delineated using segmentation. This method⁶² avoids the problem of pixels from within-crown shadows, from tree edges or from sunlit or shaded soil background affecting the spectra, as it retrieves the plant traits from pure sunlit vegetation components of the trees”.

P15 Statistical analysis: the choice was made to combine/compare the RF and SVM method: from description it's not apparent how the modelling results of both methods were used for the uncertainty assessment using the 68th percentile as threshold. The procedure for the second-stage classification process needs additional explanation: which training procedure was used here?

Thanks for raising this point. The main idea of including both Random Forest (RF) and non-linear SVM algorithms was to validate the results obtained by two of the most conventional and accurate machine learning algorithms, which have shown similar performance even compared to novel deep learning algorithms. Nevertheless, as one of the stages of our strategy involved the implementation of feature-weighted random forest algorithm to include the contribution of the selected spectral traits into the classification, we considered only the importance obtained by RF model through the OOB permutation

algorithm. To avoid confusion regarding this part, we decided to remove the SVM validation part since it provided a secondary validation rather than a fundamental step to obtain the final results. As a result, the SVM machine algorithms were only used in this study to retrieve the leaf biochemical constituents and the canopy structural parameters as described on Page 19, Lines 622-646.

We have clarified this aspect in the text as indicated below:

Page 20, lines 649-653:

“We implemented random forest (RF)⁶⁴ algorithms to classify healthy vs. infected (biotically stressed) trees, and non-stressed vs. water (i.e. abiotically) stressed trees for both tree species. RF algorithms have been widely used in remote sensing studies since they have shown excellent classification accuracies and high processing speeds with high-dimensional data⁶² and have shown to be accurate in detection of several diseases^{29,65–67}”

Regarding the question related to the uncertainty and the second stage methodology, the uncertainty was obtained by analysing the classification probabilities obtained by the feature-weighted random forest algorithm as described on page 20. The “uncertain” trees were included in the second-stage classification. This stage consisted of an unsupervised spectral clustering algorithm which is a graph-based algorithm that has been successfully used in remote sensing studies for classifying hyperspectral, high-resolution, and synthetic aperture radar data (Tasdemir et al., 2015; Zhao et al., 2019; Xia et al., 2015; Zhang et al., 2008). To clarify this, we included a paragraph as follows:

Page 20, lines 671-678:

“Probabilities of the predictions were obtained for each sample⁷¹ and the uncertain trees were assessed. To extract the uncertainty for each individual tree on the classification, we evaluated the probability distribution for each class from each dataset independently. Then, those trees with a classification probability below the 68th percentile (μ [mean] + σ [standard deviation]) were considered as uncertain and incorporated into a second-stage classification process. The second stage consisted of an unsupervised graph theory–based spectral clustering algorithm⁷² and included traits selected by focusing on the divergent biotic–abiotic stress obtained from the biotic and the abiotic stress databases. Spectral clustering was performed in R using the kernlab package⁷³.”

P20 Table 2: there seems to be disbalance between the large number of spectral indices and the SIF and thermal derived indices. When these are used as input for the ML methods, will these methods be able to highlight the importance of the individual thermal index in relation to the spectral indices.

We improved the text to address this concern. The nature of the hyperspectral imagery enables the calculation of several ratios / indices throughout the visible, near-infrared and short-wave infrared spectral regions. Nevertheless, the thermal data are broad-band images covering the 8-14 um region, therefore the only indicator that can be calculated is the surface temperature. Although there are other formulations for analysing the thermal region (WSI, WDI), these other indices have not been as extensively and robustly validated for these crops as CWSI. Nevertheless, the statistical analysis ensured that the spectral traits are not collinear, therefore reducing the number of inputs from the hyperspectral data, and accounting for SIF and CWSI (thermal) along with the rest of hyperspectral traits. Random

Forest models are particularly well-suited to cope with collinearity between predictor variables, in contrast to alternatives like generalised linear models (Breiman L. 2001, Random forests. Machine Learning). Hence, residual collinearity in the predictors based on the hyperspectral data will not skew or bias the results. This aspect has been clarified in the Main Text of the paper:

Page 5, lines 129-138:

“We obtained these results through a multilayered functional plant trait scheme²⁴ derived from the inversion of a physical radiative transfer model and a machine learning (ML) algorithm³⁰, applied here for the first time across two different host species. The numerous visible and near-infrared (VNIR) spectral indices initially calculated (Table 2, Extended Data) were reduced by a multicollinearity analysis based on the variance inflation factor (VIF). The latter enabled the enhanced contribution of the thermal trait (CWSI), the solar-induced fluorescence (SIF), and the model-estimated traits such as the leaf biochemical constituents and the canopy structural parameters on the disease detection. To make the results comparable across species and pathogens, the obtained importance for each spectral trait was normalised by the highest importance of each pathogen/species within each ML model (see Methods for detailed description).”

*** final point: next time add line numbers to the manuscript ***

Sorry for not adding the number in the first submission. The line numbers now have been added.

Reviewer #2

In this study, conventional machine learning algorithms including RF and SVM were developed to classify healthy vs. infected (biotic stress) trees, and water stressed vs. non-water stressed (abiotic stress) trees in both tree species. However, these textbook level methods are not novel, which makes the paper difficult to be published in such a high-quality journal.

Thank you for this review, although we would have appreciated more detailed comments on specific aspects for improvement. We disagree with this remark because:

1. The paper is not machine-learning-focused, as we use the machine learning algorithms to understand the spectral traits related to physiological changes of vegetation undergoing biotic and abiotic-induced stress. More specifically, we used machine learning as a tool to disentangle the divergent alterations caused by both types of stressors, therefore focusing on the physiological changes across species and pathogens.
2. The major novelty of the paper lies in showing that the effects of biotic and abiotic plant stress can be separated using imaging spectroscopy and thermography. Although several publications have shown that hyperspectral data can be used to detect disease symptoms, no study has focused on separating biotic- and abiotic-induced symptoms under natural open-field outbreak conditions, a significant breakthrough in remote sensing for plant diseases.
3. While it is not the primary focus of the paper, it is worth noting that the machine learning framework proposed in this paper has not been published before. It is based on a two-stage algorithm, and the main novelty is that we account for abiotic-induced physiological changes to improve the detection of biotic-induced symptoms. To do this, we have implemented a multi-stage procedure which, to our knowledge, has not been proposed before. This procedure couples the Feature-Weighted random forest algorithm with an unsupervised spectral clustering second stage. This procedure differs from previous methodologies since it includes the importance of the spectral traits into the classification (which only few algorithms do) and allows the unsupervised-based reclassification of the “uncertain” trees considering the abiotic-divergent traits. This framework builds on recent publications in remote sensing journals with high impact, as indicated below.
 - Thomas, V. A., Wynne, R. H., Kauffman, J., McCurdy, W., Brooks, E. B., Thomas, R. Q., & Rakestraw, J. (2021). Mapping thins to identify active forest management in southern pine plantations using Landsat time series stacks. *Remote Sensing of Environment*, 252, 112127.
 - Hornero, A., Zarco-Tejada, P. J., Quero, J. L., North, P. R. J., Ruiz-Gómez, F. J., Sánchez-Cuesta, R., & Hernandez-Clemente, R. (2021). Modelling hyperspectral-and thermal-based plant traits for the early detection of *Phytophthora*-induced symptoms in oak decline. *Remote Sensing of Environment*, 263, 112570.
 - Liu, M., Liu, J., Atzberger, C., Jiang, Y., Ma, M., & Wang, X. (2021). *Zanthoxylum bungeanum* Maxim mapping with multi-temporal Sentinel-2 images: The importance of different features and consistency of results. *ISPRS Journal of Photogrammetry and Remote Sensing*, 174, 68-86.
 - Johansen, K., Duan, Q., Tu, Y.-H., Searle, C., Wu, D., Phinn, S., Robson, A., & McCabe, M. F. (2020). Mapping the condition of macadamia tree crops using multi-spectral UAV and WorldView-3 imagery. *ISPRS Journal of Photogrammetry and Remote Sensing*, 165, 28.

- Selvaraj, M. G., Vergara, A., Montenegro, F., Ruiz, H. A., Safari, N., Raymaekers, D Ocimati, W., Ntamwira, J., Tits, L., Omondi, A. B., & Blomme, G. (2020). Detection of banana plants and their major diseases through aerial images and machine learning methods: A case study in DR Congo and Republic of Benin. *ISPRS Journal of Photogrammetry and Remote Sensing*, 169, 110-124.

In summary, we feel that the main advances of this paper are not on the machine learning component of this study but instead on the understanding of the physiological effects on spectral traits. Thus, we propose a novel framework that accounts for the abiotic status of the vegetation to detect pathogen-induced stress. Although other papers have shown success in detecting specific pathogens-induced physiological symptoms, this is the first study showing that disentangling biotic- from abiotic-induced alterations is feasible. We have specifically addressed this in the following sections of the paper:

In the Abstract:

*“Abstract. Plant pathogens pose increasing threats to global food security, causing yield losses that exceed 30% in food deficit regions. *Xylella fastidiosa* (Xf) represents the major transboundary plant pest and one of the world’s most damaging pathogens in terms of socioeconomic impact. Spectral screening methods are critical to detect non-visual symptoms of early infection and prevent spread. However, the subtle pathogen-induced physiological alterations that are spectrally detectable are entangled with the dynamics of abiotic stresses. **Here, using airborne spectroscopy and thermal scanning of areas covering more than one million trees of different species, infections and water stress levels, we reveal the existence of divergent pathogen- and host-specific spectral pathways that can disentangle biotic-induced symptoms. We demonstrate that uncoupling this biotic–abiotic spectral dynamics diminishes the uncertainty in the Xf detection to below 6% across different hosts.** Assessing these deviating pathways against another harmful vascular pathogen that produces analogous symptoms, *Verticillium dahliae*, the divergent routes remained pathogen- and host-specific, revealing detection accuracies exceeding 92% across pathosystems. These urgently needed hyperspectral methods advance early detection of devastating pathogens to reduce the billions in crop losses worldwide.”*

In the Main Text, Page 2, lines 84-87:

*“In this study, **we successfully disentangled biotic stress caused by vascular system-invading pathogens from abiotic stress imposed by water limitation by revealing distinct spectral pathways** associated with the physiological alterations detected through imaging spectroscopy and thermal data.”*

In the Main Text, Page 3, lines 100-105:

“We used these data to monitor i) how the Xf pathogen affected two different species (almond vs. olive), and ii) how one species (olive) responded to infection by two different xylem-limited pathogens (Xf vs. Vd). Our aim was to evaluate the robustness of distinct spectral traits to detect the biotic stress-induced symptoms, comparing across species and pathogens, while disentangling their specific spectral alterations from those caused by abiotic stress-induced dynamics.”

In the Main Text, Page 11, lines 309-314:

*“The work presented here **demonstrates that potentially confounding symptoms of biotic and abiotic stress can be distinguished for particular host plant species.** Our analyses of the most comprehensive high-resolution imaging dataset of pathogen-specific hyperspectral traits compiled so far **show, for the first time, the existence of host- and pathogen-specific spectral plant responses that diverge between biotic and abiotic stresses.** Our work goes beyond current knowledge, accurately detecting harmful xylem-limited pathogens across host species.”*

We hope these clarifications and the changes made in the manuscript help to understand better the contribution of this paper on the early detection of harmful diseases worldwide using advanced imaging spectroscopy technologies.

Reviewer #3

Authors analyzed over one million olive and almond trees using an airborne imaging spectroscopy and thermal imaging facility targeting infected and healthy trees. The biotic and abiotic stress datasets are valuable to the society. Authors used a feature-weighted ML algorithm to unravel these confounding spectral alterations induced by biotic and abiotic stressors. The work is novel in identifying biotic and abiotic stresses using big data and machine learning approaches but may not be of interest to broader community.

Thank you for your comments. We are convinced this study is timely and relevant to the broader community due to the increasing emergence and re-emergence of plant pathogens infecting agricultural and forestry areas worldwide. They cause yield losses exceeding 30% in regions where food security is critical. Global warming and international trade exacerbate the risks. At the same time, world food production needs to increase by 50% over the next 30 years to feed an increasing global population, despite diminishing arable land and climate change. On the other hand, the need for efficient use of water for irrigation has critical implications on the potential spread of pathogens. As indicated in the paper, the interaction between water stress and plant diseases has been demonstrated essential on disease evolution and resistance. Thus, the development of large-scale technologies for the early detection of pathogen infections is therefore vital, and will be needed for the global monitoring of plant diseases to avoid its spread into new areas and minimise yield losses. We believe that our work could also broaden the interests of a wider community working on crop protection and management strategies and across a large number of disciplines, comprising remote sensing, plant pathology, artificial intelligence, engineering and sensing technologies, space applications, drone industry, and agronomy and physiology, among others. In the final remarks of the manuscript, we have included the following paragraph to help clarify the importance of this study across disciplines and to discuss the prospects for a global monitoring system.

Pages 11-12, lines 309-340:

“The work presented here demonstrates that potentially confounding symptoms of biotic and abiotic stress can be distinguished for particular host plant species. Our analyses of the most comprehensive high-resolution imaging dataset of pathogen-specific hyperspectral traits compiled so far show, for the first time, the existence of host- and pathogen-specific spectral plant responses that diverge between biotic and abiotic stresses. Our work goes beyond current knowledge, accurately detecting harmful xylem-limited pathogens across host species.

Global warming and international trade are exacerbating risks related to emerging and reemerging pathogens threatening agriculture. At the same time, world food production needs to increase by 50% over the next 30 years to feed a growing global population, despite decreasing arable land and climate disruption⁴⁸. With yield losses due to pathogens exceeding 30% in regions where food security is critical, the development of technologies for large-scale early detection of outbreaks is crucial. A global plant disease monitoring framework will require collaboration across disciplines, including remote sensing, physics, artificial intelligence, engineering and sensor development, and space and drone industries, interacting closely with plant pathology, physiology, and agronomy.

The analytical approach introduced here provides a transferrable framework to disentangle pathogen-induced stress from abiotic dynamics in a range of species, which is critical for the

development of global disease-detection models. Detecting the coexistence of both factors is also fundamental to evaluate the evolution and treatment of the plant, either to adapt the treatment in case there is only one stress factor, or to control the interaction of biotic and abiotic stresses. For example, drought can play a key role in the development of plant diseases^{49,50}. The applicability of this framework to other pathotypes will require further considerations; for individual plant diseases, it will depend on the degree of divergence of the spectral pathways induced by the coupled biotic–abiotic stress-related physiological alterations for each species. We expect results to further improve for non-xylem-limited pathogens that cause physiological responses uncoupled from the abiotic dynamics of water stress. Widespread use will require further developments of technology readiness; a critical limitation for the operational application of these methods lies in the need for high-spatial resolution hyperspectral and thermal imaging (i.e., at sub-meter resolutions), a technology currently available only from drones at small scale, and from manned airborne platforms such as the ones used here. Future hyperspectral sensors on board satellites or high-altitude drones may enable systematic data collections with imaging spectroscopy at sub-meter resolution data, and, when combined with analytical frameworks, permit the real-time monitoring of diseases and abiotic stresses at global scales.”

I have several major concerns regarding the design of the experiments:

1. Authors used olive and almond trees as examples but these types of trees are used for different purposes. This is not easy for readers to follow and provides no significant contribution for comparison as illustrated in Figure 1. The spectral changes of olive trees are used to identify divergent Xf- and Vd-induced biotic-abiotic spectral alterations while the spectral changes of almond trees are used to identify the drivers from abiotic stress. Figure 1 is somewhat confusing with Figure 3. I would suggest to focus Xf-induced biotic-abiotic spectral alterations for both olive and almond and further analyze the conditions of irrigated or not for both trees as well.

Thank you for these comments, which helped us clarify some aspects of the paper.

The paper focuses on olive and almond species because of Xf's devastating effect on these two species in Europe and because Vd is the most harmful soilborne disease for olive trees worldwide. The results reported in this study were from actual outbreaks in different locations in Europe. The interest of reporting these results in olive and almond is that we compared: i) the same pathogen, Xf, affecting two different species (almond vs. olive), and ii) the same species (olive) affected by two xylem-limited pathogens (Xf vs. Vd). Through multiple analyses, we could assess the robustness of specific spectral traits to detect the biotic-induced symptoms, comparing across species and pathogens and accounting for the abiotic-induced stress dynamics.

We have made this aspect clearer, and a paragraph has been added in Page 3, lines 100-105:

“We used these data to monitor i) how the Xf pathogen affected two different species (almond vs. olive), and ii) how one species (olive) responded to infection by two different xylem-limited pathogens (Xf vs. Vd). Our aim was to evaluate the robustness of distinct spectral traits to detect the biotic stress-induced symptoms, comparing across species and pathogens, while disentangling their specific spectral alterations from those caused by abiotic stress-induced dynamics”.

Regarding the potential confusion between Figure 1 (new Figure 2) and Figure 3 (new Figure 4), we would like to clarify that Figure 2 shows the importance of spectral traits due to biotic-only induced effects. In contrast, Figure 4 shows the importance of spectral traits for both the biotic and abiotic-induced dynamics. In addition, Figure 4 shows the traits (indicated with ★) that yielded the highest difference for the biotic/abiotic pathways. To make this clearer, we have added titles to the figures:

- New Figure 1. High-resolution airborne hyperspectral image acquired over one of the Apulia *Xylella fastidiosa* (Xf) infected areas.
- Figure 2. Importance of spectral traits to detect Xf- and Vd-infection symptoms
- Figure 3. Importance of spectral traits to detect abiotic-induced water stress symptoms
- Figure 4. Importance of spectral plant traits for Xf- and Vd-detection across species under simultaneous biotic and abiotic stress

We hope that these changes clarify the differences between the figures.

2. The manuscript lacks clear flow of the information and results. Authors should provide an improved design of experiments to illustrate the common and unique findings to unravel confounding spectral alternations induced by biotic and abiotic stressors.

The paper describes the results obtained when biotic-only stress is assessed for Xf and Vd detection, discussing that several trees were misclassified due to the uncertainty error associated with the abiotic variability. Then the abiotic-only analysis is introduced to show that the spectral traits related to water stress are different across species and that they diverge from the ones obtained from biotic-only induced stress. Finally, by accounting for the variability of abiotic stress, we identify the distinct spectral traits that disentangle biotic-abiotic symptoms and across two species and pathogens.

This sequence is necessary to justify accounting for abiotic stress to improve the detection of biotic stress symptoms. The revised manuscript contains additional text to aid the reader through the sequence of analyses and results (the addition of titles to the figures, we hope, further helps in this respect):

Page 3, lines 100-105:

“We used these data to monitor i) how the Xf pathogen affected two different species (almond vs. olive), and ii) how one species (olive) responded to infection by two different xylem-limited pathogens (Xf vs. Vd). Our aim was to evaluate the robustness of distinct spectral traits to detect the biotic stress-induced symptoms, comparing across species and pathogens, while disentangling their specific spectral alterations from those caused by abiotic stress-induced dynamics”.

Page 8, lines 209-212:

“Our feature-weighted models, which at this stage account only for the biotic stress-induced variability in the spectral traits yielded OA=84% ($\kappa=0.68$) for Xf detection of infection in almond ($n=4,048$ trees), and OA=77% ($\kappa=0.43$) and OA=75% ($\kappa=0.49$) for Xf and Vd detection of infection in olive, respectively ($n=7,296$ and $n=1,852$ trees, respectively).”

Page 9, lines 228-234:

“This multitemporal dataset enabled the identification of individual trees that showed consistent and sustained water status levels across seasons. The temporal approach allowed clustering of trees with different stress levels over a long period of time (i.e., two growing seasons) rather than focusing the analysis on short-term water stress conditions potentially due to transitory environmental effects or irrigation system malfunctions in each orchard under study. Thus, the multiyear dataset improved the selection of trees consistently experiencing sustained water stress.”

3. Detailed data illustration for each experiment should be provided to better understand the group of samples. Authors should provide further analyses and discussions on whether different areas and years would impose additional divergent pathways.

We kindly refer here to the Methods section, particularly under the Field data collection subheading, which provides details on sampling design across sites, years, and disease infection status:

Pages 16-18, lines 507-561:

“1) Xf and Vd biotic stress dataset. Field assessments of Xf- and Vd-infected trees were carried out from outbreaks affecting olive and almond species in the indicated regions of Italy and Spain between 2011 and 2019^{24,35,52}. During these campaigns, we performed quantitative PCR (qPCR)⁵⁵ for Xf in olive and almond (Alicante), recombinase-polymerase-amplification (RPA) using the AmplifyRP XRT+ test (Agdia®, Inc., Elkhart, IN)⁵⁶ for Xf in almond (Majorca) or conventional PCR⁵⁷ assays for Vd, as well as visual assessments in individual trees of disease incidence (DI) and disease severity (DS). DS was scored using a 0–4 rating scale according to the percentage of the tree crown showing disease symptoms.

In Apulia, the Xf-olive database comprised a total of 15 olive groves surveyed during the June 2016 and July 2017 campaigns. Visual assessments for infection were conducted on 7,296 trees (3,324 in 2016 and 3,972 in 2017). In 2016, 1,886 symptomatic (and 1,438 asymptomatic) trees were surveyed (762 trees labelled as DS=1; 802 DS=2; 250 DS=3 and 72 DS=4). In 2017, 1,365 were reported as symptomatic (and 2,607 asymptomatic) (686 DS=1; 542 DS=2; 122 DS=3 and 15 DS=4). qPCR assays were carried out to diagnose Xf infection in 77 olive trees, whereby 39 trees tested negative (qPCR=0) and 38 tested positive (qPCR=1).

On the island of Majorca and at the Alicante province, the field-based Xf-almond database comprised a total of 19 almond groves surveyed in 2018 and 2019, respectively. In Alicante, the field surveys covered 83 ha with 9 almond groves consisting of 943 almond trees. During the field campaigns, almond trees were visually assessed to evaluate Xf-induced DI and DS indices. From this analysis, we identified 593 symptomatic trees and 350 asymptomatic trees. Out of all symptomatic trees, 163 were rated as DS=1, 214 DS=2, 157 DS=3, and 59 DS=4. Furthermore, qPCR analysis was carried out on 226 almond trees to diagnose Xf infection, resulting in 48 non-infected (qPCR=0) almond trees and 178 infected trees (qPCR=1). In Majorca, field surveys in July 2019 covered a total of 2,803 ha and comprised 10 almond groves. During the field campaigns, visual observations were carried out on over 4,048 almond trees to assess DI and DS, yielding 1,387 symptomatic and 2,661 asymptomatic trees. From symptomatic trees, 537 were rated as DS=1, 449 DS=2, 359 DS=3, and 42 DS=4. We conducted AmplifyRP XRT+ assays on 265 almond

trees for diagnosing Xf infection the same day they were sampled and identified 141 negative trees (qPCR=0) and 124 positive trees (qPCR=1).

...

We assessed Vd-infected olive trees from 11 olive groves by surveying an area of over 3,000 ha in Castro del Rio and Ecija, southern Spain, in 2011 and 2013, respectively. In Castro del Rio, we conducted visual assessments in an infected area of 96 ha comprising 1,878 olive trees, thus identifying 1,569 asymptomatic and 283 symptomatic olive trees. Out of the 283 symptomatic trees, 218 were rated as DS=1; 45 DS=2; 12 DS=3 and 8 DS=4. We measured leaf Fs and Fm' fluorescence parameters from 25 leaves per tree using a PAM-2100 Pulse-Amplitude Modulated Fluorometer (Heinz Walz GMBH, Effeltrich, Germany). In addition, leaf PRI570 was measured from 25 leaves per tree using a custom-made PlantPen device (Photon System Instrument, Drasov, Czech Republic). Finally, we measured leaf conductance (Gs) on five leaves per tree using a leaf porometer (model SC-1, Decagon Devices, Washington, DC, USA). In the Écija region, the surveyed area covered 3,424 ha, and 5223 olive trees were evaluated. We performed visual assessment to determine DI and DS indices of Vd-infected trees, identifying 5,040 asymptomatic olive trees. Of the remaining 183 olive trees that were symptomatic, 112 were trees rated as DS=1; 41 DS=2; 22 DS=3 and 8 DS=4."

Regarding the discussion to provide further analyses on whether different areas and years would impose additional divergent pathways, we have added a discussion about the limitations of this method when discussing the final remarks of the manuscript:

Page 11-12, lines 309-340:

"The work presented here demonstrates that potentially confounding symptoms of biotic and abiotic stress can be distinguished for particular host plant species. Our analyses of the most comprehensive high-resolution imaging dataset of pathogen-specific hyperspectral traits compiled so far show, for the first time, the existence of host- and pathogen-specific spectral plant responses that diverge between biotic and abiotic stresses. Our work goes beyond current knowledge, accurately detecting harmful xylem-limited pathogens across host species.

...

The analytical approach introduced here provides a transferrable framework to disentangle pathogen-induced stress from abiotic dynamics in a range of species, which is critical for the development of global disease-detection models. Detecting the coexistence of both factors is also fundamental to evaluate the evolution and treatment of the plant, either to adapt the treatment in case there is only one stress factor, or to control the interaction of biotic and abiotic stresses. For example, drought can play a key role in the development of plant diseases^{49,50}. The applicability of this framework to other pathotypes will require further considerations; for individual plant diseases, it will depend on the degree of divergence of the spectral pathways induced by the coupled biotic–abiotic stress-related physiological alterations for each species. We expect results to further improve for non-xylem-limited pathogens that cause physiological responses uncoupled from the abiotic dynamics of water stress. Widespread use will require further developments of technology readiness; a critical limitation for the operational application of these methods lies in the need for high-spatial resolution hyperspectral and thermal imaging (i.e., at sub-meter resolutions), a technology currently available only from drones at small scale,

and from manned airborne platforms such as the ones used here. Future hyperspectral sensors on board satellites or high-altitude drones may enable systematic data collections with imaging spectroscopy at sub-meter resolution data, and, when combined with analytical frameworks, permit the real-time monitoring of diseases and abiotic stresses at global scales.”

4. Authors claimed “hyperspectral methods advance early detection of devastating pathogens” in the abstract. The current manuscript, however, does not provide sufficient support on in-season detection and lacks on discussion of different contributions by hyperspectral and thermal images. Authors should provide further analyses on divergent contributions of hyperspectral and thermal images.

Thank you for this comment. In-season detection using time series of airborne hyperspectral data and field *re-visits* for visual inspections and qPCR analyses were reported in Zarco-Tejada *et al.* (2018) showing that the false positives detected by remote sensing developed symptoms a few months later with higher probability than the rest of the monitored trees. This work demonstrated that remote sensing could detect pre-visual symptoms before plant pathologists in the field. The paper is cited in this manuscript. Below we provide the excerpts, from both text and figures, of the revised manuscript that address the divergent contributions of hyperspectral and thermal images, particularly discussing the hyperspectral traits, fluorescence and thermal.

Page 5, lines 114-128:

“Our analysis of high-resolution airborne hyperspectral and thermal images collected over Vd (Figure 2a) and Xf (Figure 2b,c,d) outbreaks showed that infection-induced physiological alterations led to changes in biotic stress-sensitive spectral traits that were common between host species, while other traits deviated between the plant species and appeared to be host-specific (Figure 2b vs. c). In both host species, Xf infection altered spectral plant traits related to stomatal conductance dynamics as the infection progressively blocked xylem vessels and thus reduced transpiration³⁶. Lower transpiration rates also raise the overall tree canopy temperature, as measured by the thermal crop water stress index, CWSI³⁷, which is accompanied by a reduction in photosynthesis observed through solar-induced fluorescence emission signal (SIF), and alterations in the dynamics of the xanthophyll pigment cycle (for which PRIn provides a proxy) (see Table 2, Extended Data for a complete list of spectral plant traits).”

Page 5, lines 129-138:

“We obtained these results through a multilayered functional plant trait scheme²⁴ derived from the inversion of a physical radiative transfer model and a machine learning (ML) algorithm³⁰, applied here for the first time across two different host species. The numerous visible and near-infrared (VNIR) spectral indices initially calculated (Table 2, Extended Data) were reduced by a multicollinearity analysis based on the variance inflation factor (VIF). The latter enabled the enhanced contribution of the thermal trait (CWSI), the solar-induced fluorescence (SIF), and the model-estimated traits such as the leaf biochemical constituents and the canopy structural parameters on the disease detection. To make the results comparable across species and pathogens, the obtained importance for each spectral trait was normalised by the highest importance of each pathogen/species within each ML model (see Methods for detailed description).”

Figure 1 (new Figure 2) caption:

***“Fig. 2 | Importance of spectral traits to detect Xf- and Vd-infection symptoms. a–d Normalised importance of hyperspectral and thermal plant traits retrieved from the pool of spectral plant traits used to detect Verticillium dahliae, Vd- (a) and Xylella fastidiosa, Xf-induced infection symptoms (b,c,d) across olive (a,b) and almond (c,d) trees. For reference, the full list of spectral plant traits is available in the Extended Data, Table 2.
...”***

Figure 2 (new Figure 3) caption:

***“Fig. 3 | Importance of spectral traits to detect abiotic-induced water stress symptoms. Sensitivity of plant spectral traits calculated from hyperspectral and thermal imagery from trees under increasing abiotic stress (caused by decreasing water stress levels, from S1 to S4) across olive (a) and almond (b) trees.
...”***

Figure 3 (new Figure 4) caption:

***“Fig. 4 | Importance of spectral plant traits for Xf- and Vd-detection across species under simultaneous biotic and abiotic stress. Spectral plant traits that diverge under biotic and abiotic stress are indicated with * for Xylella fastidiosa (Xf) infection, in olive (a) and almond trees (b,c) and for Verticillium dahliae (Vd) infection in almond trees (d).
...”***

We would also like to clarify that CWSI was defined in the Main Text as an indicator calculated from the thermal region and that Table 2 provides the complete list of spectral indicators used in this study, organised by functional groups and spectral regions:

Page 5, lines 120-124:

“Lower transpiration rates also raise the overall tree canopy temperature, as measured by the thermal crop water stress index, CWSI³⁷, which is accompanied by a reduction in photosynthesis observed through solar-induced fluorescence emission signal (SIF), and alterations in the dynamics of the xanthophyll pigment cycle (for which PRI_n provides a proxy) (see Table 2, Extended Data for a complete list of spectral plant traits).”

Minor comments:

1. Citations should be removed in the abstract.

Thank you, citations have been removed from the Abstract.

2. The limitations and further implications of the study are not well written.

Thank you for raising this issue. We have rewritten the final remarks following the comments from all reviewers. We hope these are better described and clearer.

Pages 11-12, lines 309-340:

“The work presented here demonstrates that potentially confounding symptoms of biotic and abiotic stress can be distinguished for particular host plant species. Our analyses of the most comprehensive high-resolution imaging dataset of pathogen-specific hyperspectral traits compiled so far show, for the first time, the existence of host- and pathogen-specific spectral plant responses that diverge between biotic and abiotic stresses. Our work goes beyond current knowledge, accurately detecting harmful xylem-limited pathogens across host species.

Global warming and international trade are exacerbating risks related to emerging and reemerging pathogens threatening agriculture. At the same time, world food production needs to increase by 50% over the next 30 years to feed a growing global population, despite decreasing arable land and climate disruption⁴⁸. With yield losses due to pathogens exceeding 30% in regions where food security is critical, the development of technologies for large-scale early detection of outbreaks is crucial. A global plant disease monitoring framework will require collaboration across disciplines, including remote sensing, physics, artificial intelligence, engineering and sensor development, and space and drone industries, interacting closely with plant pathology, physiology, and agronomy.

The analytical approach introduced here provides a transferrable framework to disentangle pathogen-induced stress from abiotic dynamics in a range of species, which is critical for the development of global disease-detection models. Detecting the coexistence of both factors is also fundamental to evaluate the evolution and treatment of the plant, either to adapt the treatment in case there is only one stress factor, or to control the interaction of biotic and abiotic stresses. For example, drought can play a key role in the development of plant diseases^{49,50}. The applicability of this framework to other pathotypes will require further considerations; for individual plant diseases, it will depend on the degree of divergence of the spectral pathways induced by the coupled biotic–abiotic stress-related physiological alterations for each species. We expect results to further improve for non-xylem-limited pathogens that cause physiological responses uncoupled from the abiotic dynamics of water stress. Widespread use will require further developments of technology readiness; a critical limitation for the operational application of these methods lies in the need for high-spatial resolution hyperspectral and thermal imaging (i.e., at sub-meter resolutions), a technology currently available only from drones at small scale, and from manned airborne platforms such as the ones used here. Future hyperspectral sensors on board satellites or high-altitude drones may enable systematic data collections with imaging spectroscopy at sub-meter resolution data, and, when combined with analytical frameworks, permit the real-time monitoring of diseases and abiotic stresses at global scales.”

REVIEWERS' COMMENTS

Reviewer #1 (Remarks to the Author):

The authors have made a careful and complete update of the manuscript based on the points raised in the review(s) with special attention to an improved description on several technical details and updated figures.

Based on this the manuscript could be accepted for publication.